# Hot springs viruses at Yellowstone National Park have ancient origins and are adapted to thermophilic hosts
L. Felipe Benites[1], Timothy G. Stephens[1], Julia Van Etten [1,2], Timeeka James[1], William C. Christian [3], Kerrie Barry [4], Igor V. Grigoriev [4,5], Timothy R. McDermott[6] & Debashish Bhattacharya [1] ✉

Geothermal springs house unicellular red algae in the class Cyanidiophyceae that dominate the microbial biomass at these sites. Little is known about host-virus interactions in these environments. We analyzed the virus community associated with red algal mats in three neighboring habitats (creek, endolithic, soil) at Lemonade Creek, Yellowstone National Park (YNP), USA. We find that despite proximity, each habitat houses a unique collection of viruses, with the giant viruses, Megaviricetes, dominant in all three. The early branching phylogenetic position of genes encoded on metagenome assembled virus genomes (vMAGs) suggests that the YNP lineages are of ancient origin and not due to multiple invasions from mesophilic habitats. The existence of genomic footprints of adaptation to thermophily in the vMAGs is consistent with this idea. The Cyanidiophyceae at geothermal sites originated ca. 1.5 Bya and are therefore relevant to understanding biotic interactions on the early Earth.

Extreme environments such as acidic geothermal springs (hot springs) are thought to have been "cradles of microbial life" on the early Earth[1]. In contrast, on the modern Earth, these sites represent biology at the extremes, comprising isolated "islands", usually surrounded by mesophilic life[2]. A key question is whether species that currently inhabit hot springs have anciently derived adaptations that can provide insights into primordial life. Specialized archaeal and bacterial taxa thrive and dominate hot springs, whereas eukaryotes are often absent[3,4]. However, in more moderate downstream habitats, polyextremophilic unicellular Rhodophyta (red algae) that split from their mesophilic sister taxa ca. 1.5 billion years ago[5] form dense, conspicuous mats overshadowing prokaryotic life. And, where there is life, there are viruses. Although in lower densities than in other environments[5,6], viruses appear to play a pivotal role in microbial host mortality in other hot springs, often being the only infectious agents in these sites[7,8]. Bacterial and archaeal viruses are abundant[8–10], however less is known about eukaryotic viruses, with some exceptions[11–14]. And whether eukaryotic viruses modulate adaptation to these extreme environments by reprogramming host metabolism and/or through interdomain gene transfer is currently an open question[15].

Thermoacidophilic red algae include the genera *Cyanidioschyzon* and *Galdieria* (class Cyanidiophyceae), which, in hot springs[16] such as Yellowstone National Park (YNP), detoxify arsenic and other heavy metals[17]. The major driver of polyextremophilic adaptation in these algae is horizontal gene transfer (HGT) from prokaryotic donors[5,18–20], although there exists some evidence for viral HGTs[21,22]. Thus far, no viruses infecting these algae have been identified. Microbial mats generally comprise lineages that are phylogenetically worlds apart, whereby related viral groups may interact with unrelated hosts[23]. Furthermore, microbial mat viruses may act as drivers of mat formation by triggering regeneration due to nutrient fluxes from host mortality[24,25].

The "hot start" hypothesis posits that early life flourished in a hot environment and later adapted to the cooling Earth[26], which is supported by the ancient splits of thermophilic organisms in gene trees[27–29]. Therefore, modern hot springs are ideal systems for investigating adaptations that could have been present in extreme environments in the early Earth. Specifically, longer-term infections within the mats may have led to thermal signatures of ancient associations in viral genomes (Hypothesis [H] 1). Alternatively, these environments may have been invaded more recently by

[1]Department of Biochemistry and Microbiology, Rutgers, The State University of New Jersey, New Brunswick, NJ 08901, USA. [2]Graduate Program in Ecology and Evolution, Rutgers, The State University of New Jersey, New Brunswick, NJ 08901, USA. [3]Department of Land Resources and Environmental Sciences, Montana State University, Bozeman, Montana, USA. [4]U.S. Department of Energy Joint Genome Institute, Lawrence Berkeley National Laboratory, Berkeley, CA 94720, USA. [5]Department of Plant and Microbial Biology, University of California Berkeley, Berkeley, CA, 94720, USA. [6]Department of Chemistry and Biochemistry, Montana State University, Bozeman, Montana, USA. ✉e-mail: dbhattac@rutgers.edu

viruses from mesophilic habitats (H2) and lack signatures of thermophily. To discriminate between these two hypotheses, we investigated viruses infecting red algal mats in a hot spring environment with the over-arching goals of characterizing viral community composition, elucidating local adaptation and potential role in cellular communities, and understanding virus evolutionary history. We analyzed metagenomic data from three adjacent environments at Lemonade Creek, YNP (Fig. 1a): a creek fed by an acidic hot spring, neighboring endoliths, and acidic soils (Fig. 1b). Their taxonomic diversity, phylogenetic position, and landmark genomic signatures suggest ancient and persistent adaptations by YNP viruses to geothermal habitats.

## Results

### Overview of virus community composition at Lemonade Creek, YNP

From 12 metagenomic samples spanning the three environments, we assembled 7,886,883 scaffolds of which 6,390 were predicted to be viral. After scaffold dereplication and clustering (see methods), following standard procedures for the identification of viral operational taxonomic units (vOTUs)[30,31], we found 3679 vOTUs that encode 17,102 proteins. To reconstruct larger and more complete genomes, we binned these vOTU scaffolds and recovered 25 viral metagenome-assembled genomes (vMAGs), ranging from high ($n = 5$), medium ($n = 14$), to low completeness ($n = 6$) according to CheckV[32], with lengths ranging from 1.63 kb to 2.2 Mb (Supplementary Data 1, Supplementary Data 2).

### Virus taxonomy, distribution, and abundance at Lemonade Creek

Of the 3679 vOTUs, the majority were from the class Megaviricetes ($n = 890$), eukaryotic DNA viruses that infect algae, other protists, and some metazoans[33]. Other frequent classes were Caudoviricetes ($n = 711$), containing bacterial and archaeal viruses, Pokkesviricetes ($n = 70$) that infect Metazoa and dinoflagellate algae[33], Maveriviricetes ($n = 61$) that are "virophages", parasites from giant eukaryotic viruses[34], and Tectiliviricetes ($n = 27$), associated with Bacteria, Archaea, and metazoan hosts[35]. However,

51.05% of the vOTUs ($n = 1878$) were unclassified, suggesting the presence of many diverged viruses (Supplementary Data 1).

To compare vOTU abundance profiles across samples, we calculated RPKM (Reads Per Kilobase Million) values for viral sequences at the class level (Fig. 1b). This analysis revealed that Megaviricetes (69.9%) and Caudoviricetes (14.15%) are most abundant, followed by Unclassified viruses (12.85%) that are present in all the studied environments (although the latter group are more abundant in soil samples), whereas all others represent ~1% or less, and are specific to environments. Maveriviricetes, although only comprising 0.5% of the total vOTU count, are present in all environments, which is expected given the high abundance of their known viral hosts (Megaviricetes). It is noteworthy that the most abundant viral classes observed at these sites are known to infect eukaryotic hosts.

To study viral diversity and abundance and their relationship to the environment, we performed alpha and beta diversity analyses of vOTUs at the class level. Soil has the highest diversity, whereas the creek has the lowest diversity, overall richness, and Shannon diversity among the environments (Fig. 1d). Non-metric multidimensional scaling (NMDS) (p(perm) = 0.002) analysis of beta diversity from each metagenome sample shows that samples grouped according to environment (Fig. 1c). However, the endolithic site shows higher variation of the viral community, whereby samples are positioned at different extremes of the NMDS2 axis. Endolithic samples also contain most viral classes, whereas, in the creek, samples are more tightly clustered, driven by the dominance of the Megaviricetes. This suggests that despite being adjacent, the studied environments contain unique distributions of viral classes, with the endolithic environment harboring a high and non-homogeneous diversity of viruses. This may be explained by endoliths containing unique internal microhabitats, which are better protected from low pH, high temperature, and the damaging UV irradiation associated with the other two environments in YNP (i.e., the park has an average elevation of ca. 8000 ft) [https://www.nps.gov].

### Functional profile of the Lemonade Creek virus community

Of the 17,102 proteins encoded by the 3679 vOTUs, 15.7% have assigned functions based on GO (gene ontology), KEGG (Kyoto Encyclopedia of

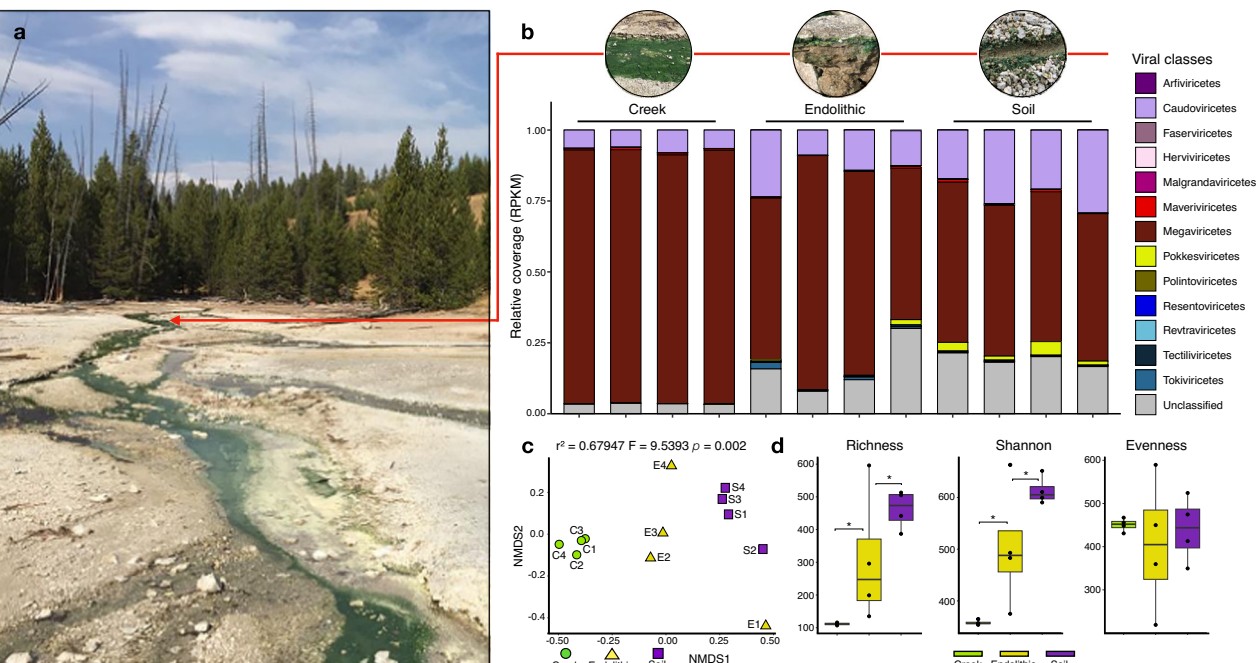

**Fig. 1 | Analysis of viruses in an extreme environment. a** Lemonade Creek, Yellowstone National Park (YNP) and the virus community at the sampled sites (see red arrow). **b** Abundance of virus classes (legend shown at right) reflecting relative coverage (reads per kilobase million, RPKM) at the creek, endolithic, and soil sites with four replicates/site. **c** Beta diversity was calculated using a non-metric multidimensional scaling (NMDS) ordination of virus classes in the creek (green circles), endolithic (yellow triangles), and soil (purple squares) communities, based on Bray–Curtis dissimilarity. **d** Alpha diversity calculated using the richness (number of virus classes), Shannon diversity, and evenness of virus communities in the three environments. The asterisks show significant statistical differences between groups.

Genes and Genomes), and PFAM (protein families) analysis (Fig. 2a–c, respectively) (Supplementary Data 3). We analyzed the top 10 annotations for each gene to investigate their putative functions. Proteins encoding cellular metabolic processes, viral genome replication, nitrogen metabolism, DNA and protein binding, and transferase activity are most frequent, although the soil contained a larger number of proteins encoding host cellular and intracellular components, as well as symbiotic and interspecies interactions. Pathways involved in RNA polymerase, DNA repair, purine and pyrimidine metabolism, drug metabolism, and homologous recombination, and, curiously, cellular structural development are the most frequent. Regarding PFAM domains, the most frequent domains are associated with viral replication, ATPases, glycosyltransferases, HNH (histidine and asparagine domain) endonucleases and protein kinases, and those involved in viral structures such as major capsid proteins (MCPs) and genome maintenance *via* helicases.

To explore the proteins with unknown functions ($n = 14,415$) (Supplementary Data 4), we retrieved protein matches to build a sequence similarity network (SSN) containing viral protein clusters (vPCs). To reduce the total protein space, we first clustered all unknown sequences with their matches in viral databases at the threshold of 60% identity and 80% coverage, which resulted in 13,543 proteins. These vPCs allowed us to transfer putative functions to a fraction of unknown proteins and evaluate their distribution and composition from the sampled environments. After removing singletons and poorly connected clusters (degrees < 8), 86 vPCs remained (Fig. 2d), with just four of the largest vPCs being functionally classifiable. The largest (vPC1) is dominated by proteins encoding DNA primase/helicase, or the origin of replication binding. vPC2 comprises homologs of homing endonucleases. In vPC3, we find a single sequence encoding a transposase with the remainder having unknown functions, whereas vPC4 contains homologs encoding restriction endonucleases. The

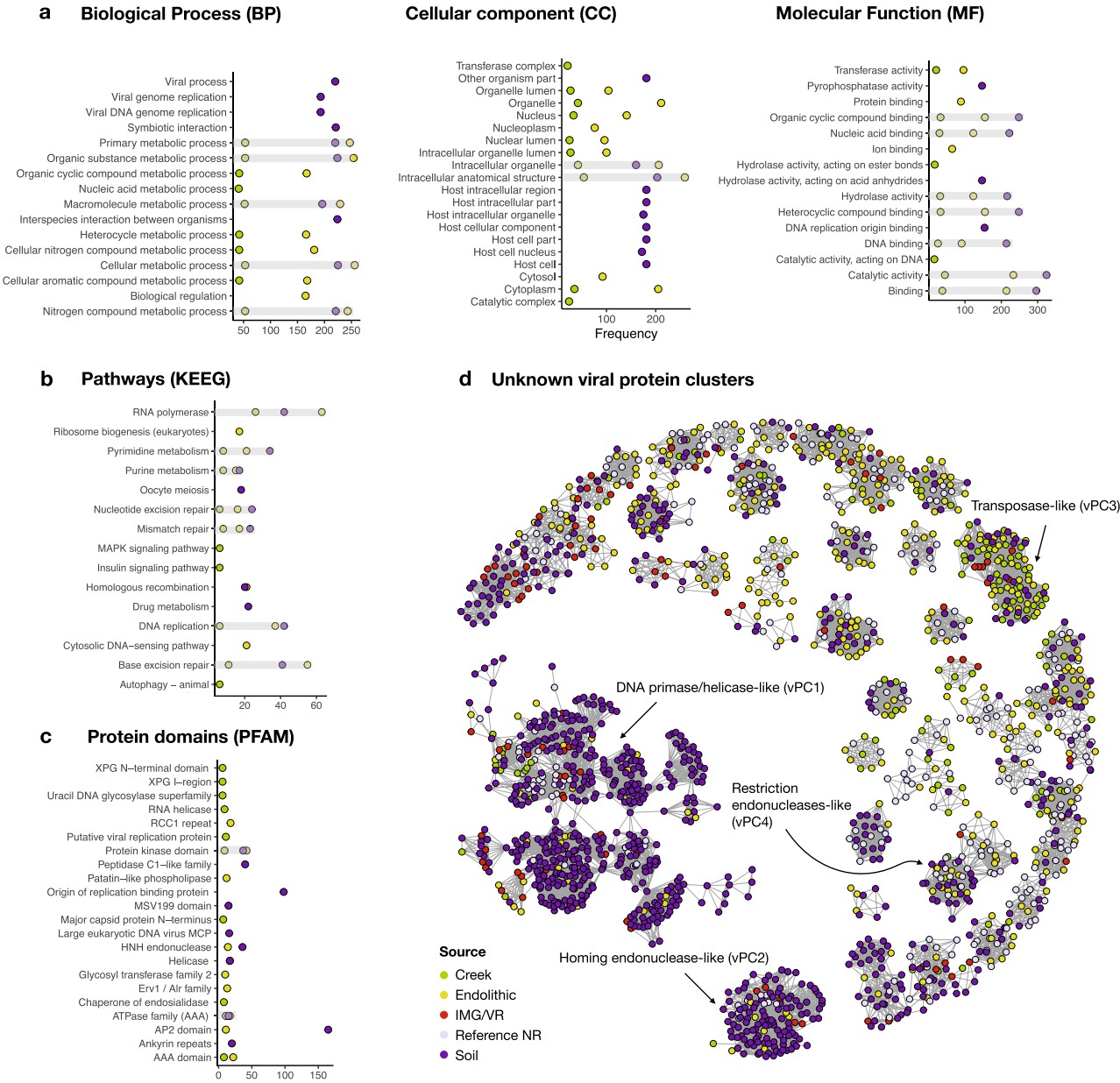

**Fig. 2 | Functions encoded by virus genomes at YNP, considering the top 10 most frequent annotations. a** Gene ontologies (GO terms) describing the categories for biological process (BP), cellular component (CC), and molecular function (MF); **b** KEGG pathways (Kyoto Encyclopedia of Genes and Genomes); **c** protein families (PFAM) according to InterPro database. **d** Sequence similarity networks containing 86 unknown viral protein clusters (vPCs). The color key shows the sampled site: creek (green), endolithic (yellow), and soil (purple), IMG/VR (Integrated Microbial Genomes and uncultivated viruses) (red), Reference/NR (lavender) denotes GenBank viral RefSeq and non-redundant databases.

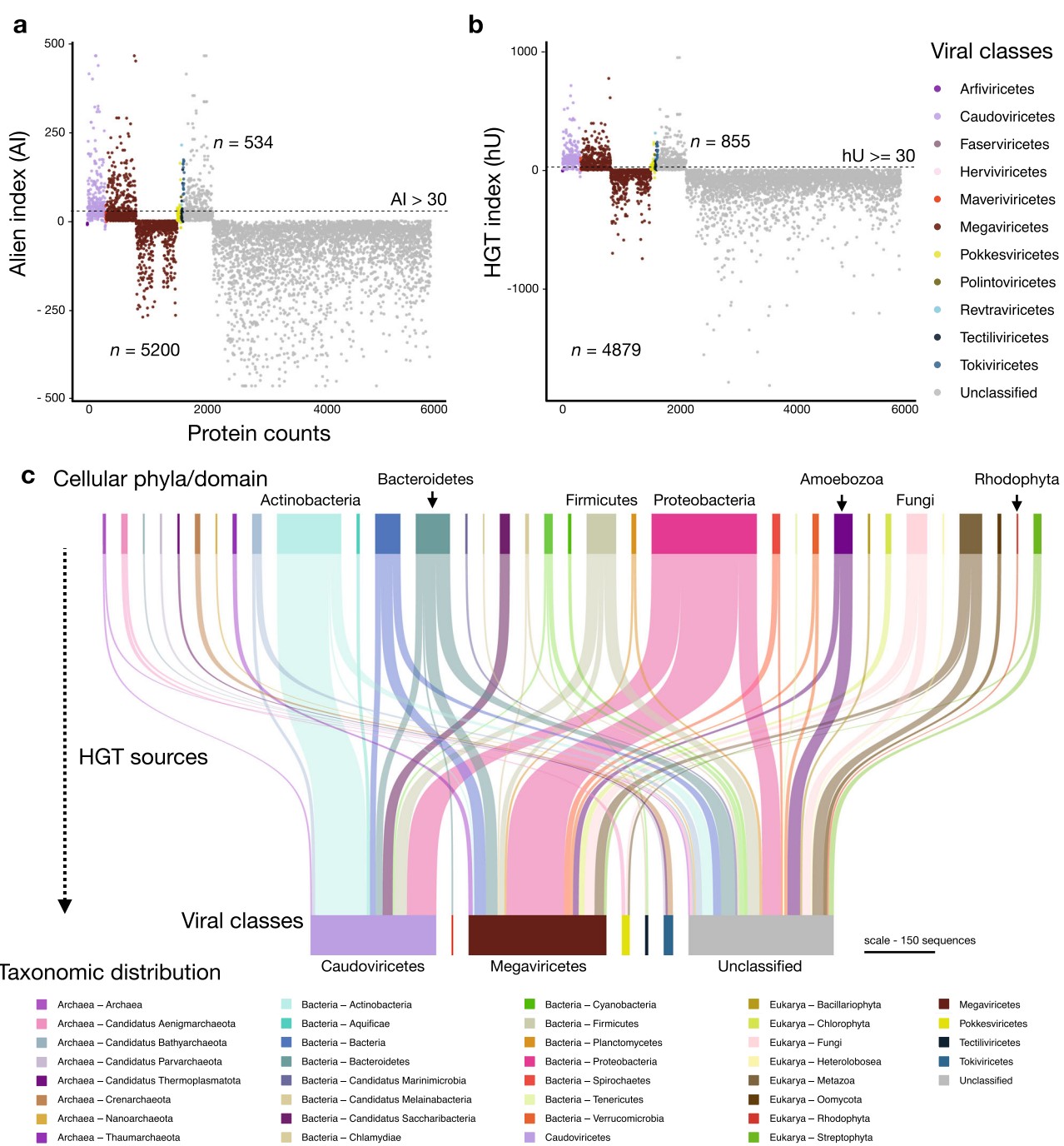

**Fig. 3 | Horizontal gene transfer (HGT) using virus operational taxonomic units (vOTUs) at the class level. a** HGT measured using the alien index (AI). **b**, HGT measured using the HGT index (hU). Proteins with indices greater than the threshold (dashed lines at AI > 30 and hU ≥ 30) are vHGT candidates. The key at right shows virus classes. **c** Sankey diagram showing the cellular sources of HGTs (top) for each virus class (bottom). The key on the bottom shows archaeal, bacterial, eukaryotic, and viral taxonomic groups and is ordered alphabetically in the same direction as the Sankey diagram. The scale bar represents the amount of putative HGT-derived genes.

remaining 82 vPCs lack annotated functions. These results suggest that hot springs harbor many uncharacterized proteins that may play specific ecological roles in each environment.

## HGT in the Lemonade Creek virus community

We investigated the impact of HGT on generating novelty (i.e., to the best of our knowledge) in Lemonade Creek viruses and explored their potential role as gene transfer agents between cellular taxa *via* viral transduction. HGTs can also provide clues of past and present virus-host associations[36]. For this

aim, we evaluated the proportion of protein sequences with cellular homologs for each vOTU and calculated their alien and HGT indices (AI and hU) (Fig. 3a, b, respectively). We identified 921 unique HGT candidates (534 using AI and 855 using hU) from 5734 sequences (representing 16% of this total) with hits to genes in cellular taxa (Fig. 3c). Bacteria was the major HGT source ($n = 658$), followed by Eukarya ($n = 196$), and Archaea ($n = 67$). We found evidence of virus-host associations. In Revtraviricetes, a class that infects Fungi[37], all HGTs ($n = 2$) are from fungal donors, in Tokiviricetes, which infect Archaeal hosts[38], all HGTs acquisitions are from

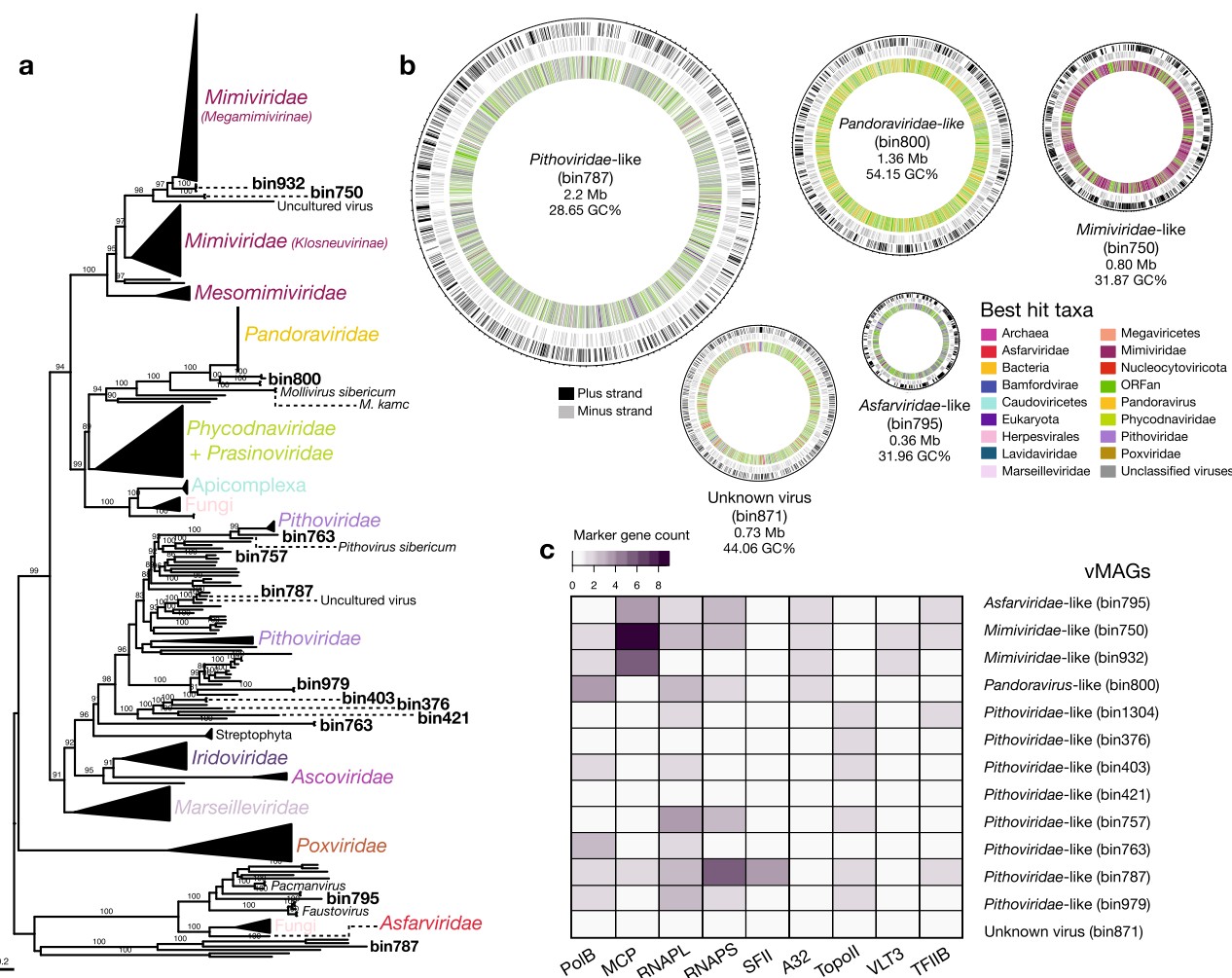

**Fig. 4 | Overview of nucleocytoviricota-related metagenome-assembled genomes (vMAGs). a** Maximum likelihood phylogenetic tree of the marker protein, B DNA Polymerase (PolB), from the vMAGs and reference viral genomes, with bootstrap support values ≥ 90% shown. **b** Circular vMAG representation with approximate sizes of selected viruses shown in the first and second outer rings (black and gray lines, respectively), the third (inner) ring is the taxonomic affiliation of the best BLAST hit. **c** Marker gene presence/absence in the vMAGs.

Archaea, and an intriguing result in the unclassified viruses, 3 HGTs (one hypothetical protein and two GDP-D-mannose-3′,5′-epimerase) are from *Galdieria* (Cyanidiophyceae) (Supplementary Data 5). Most HGT-derived cellular proteins are hypothetical/uncharacterized/unnamed (*n* = 512), followed by those in the TIGR03118 family (*n* = 13), which has a C-terminal putative exosortase interaction domain (InterPro id: TIGR03118) with unknown function, and DUF2341 domain-containing proteins (*n* = 11), which are found in bacterial proteins such as proton channels and transport proteins (InterPro id: IPR018765). Other proteins contain LysM domains (*n* = 8) (InterPro id: IPR018392), used by plants for pathogen defense and symbiosis and by some bacteriophages to degrade bacterial cell walls[39]. Other proteins, such as DNA methyltransferases (*n* = 7), phage portal proteins (*n* = 7), which may be remnants of lysogenic viruses, and timeless family proteins (*n* = 7) that regulate eukaryotic circadian rhythms, stress responses, and may be responsible for maintenance of viral latency[40,41], are less frequent. Although these results provide a lower bound for HGTs, given the conservative measures we used, the finding of bacterial genes in eukaryotic vMAGs, and vice versa, suggests that viruses act as HGT vectors in hot spring mats, implicating a role in interdomain gene acquisition, and potentially host adaptations.

## Viral metagenome-assembled genomes (vMAGs)

The 25 vMAGs produced from binning of the vOTUs were grouped into five viral families: Circoviridae (*n* = 2), Mimiviridae (*n* = 2),

Pandoraviridae (*n* = 1), Pithoviridae (*n* = 7), and Tectiviridae (*n* = 1) (Supplementary Data 2). However, 12 bins were initially unclassified. The vMAGs have varied completeness, with five having 100% completeness, 14 medium, and 6 low. The largest vMAG (bin787; Pithoviridae-like) is of size 2.2 Mb, followed by bin800 (Pandoravirus-like) of size 1.36 Mb (Fig. 4b), and the smallest ranged from 7.8 kb (bin1029) to 1.6 kb (bin901). We searched for viral markers and found 11 vMAGs that contain putative capsid and replication-associated proteins (rep), characteristic of CRESS DNA viruses (Eukaryotic Circular Rep-Encoding Single-Stranded virus). A vMAG (bin250) assigned to Tectiviridae encoded a major capsid protein and a terminase large subunit, among other proteins, confirming its initial classification. We were unable to confidently classify one vMAG (bin871), that contained hits with both unclassified viruses (Pacmanvirus, Faustovirus), Nucleocytoviricota and Caudoviricetes viruses. Although there were no major hallmark genes from the database "giant virus orthologous groups" (GVOGs), we found highly duplicated "Nucleo-Cytoplasmic Virus Orthologous Groups" (NCVOGs database) genes, as "Superfamily II helicase related to herpesvirus replicative helicase" (*n* = 96) and HNH endonuclease (*n* = 87) (Supplementary Data 6). CheckV did not identify contamination in this vMAG, therefore we could not determine if it is a novel (i.e., to the best of our knowledge) chimeric virus, like Mirusviruses[42] (despite no significant hits with these viruses) or a database primary misannotation.

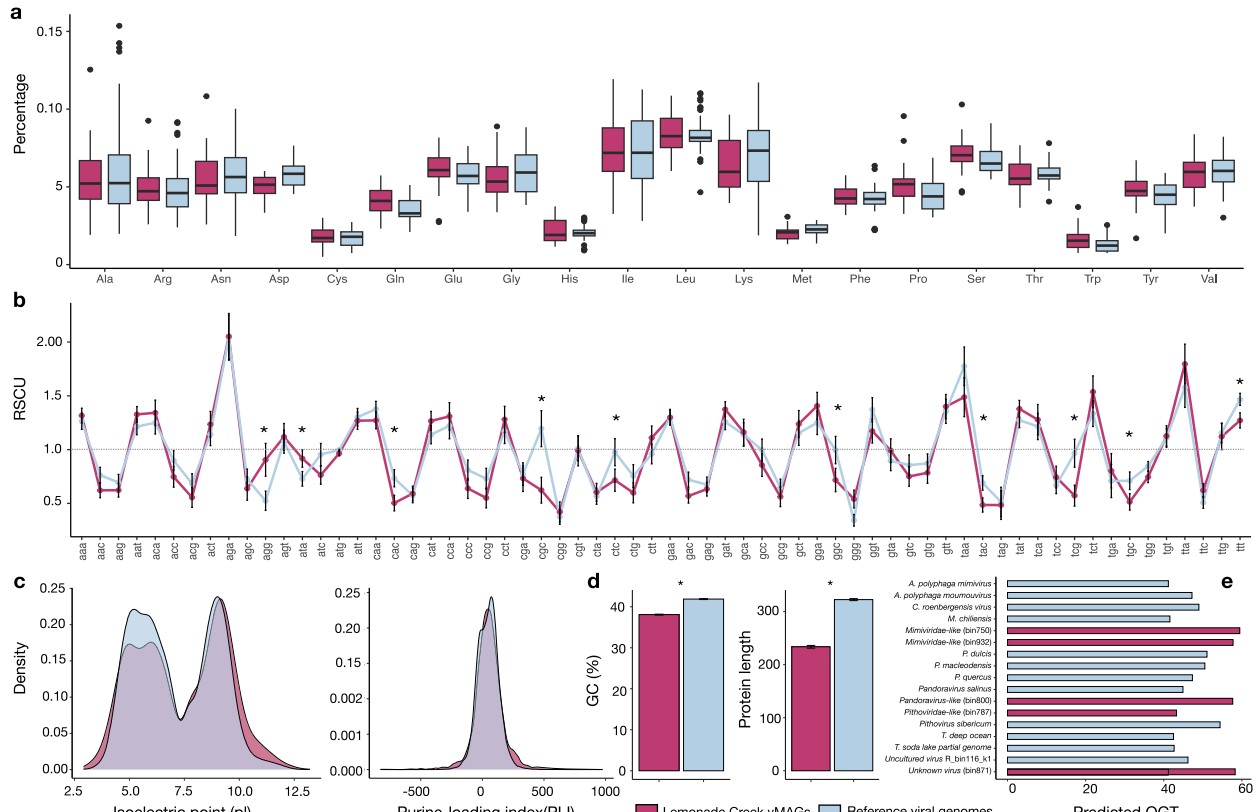

**Fig. 5 | Comparison of genome and proteome properties between Lemonade Creek hot springs virus metagenome-assembled genomes (vMAGs) (pink color) and reference virus genomes from closely related phylogenetic groups (blue color). a** Analysis of differences in amino acid frequencies. **b** Analysis of relative synonymous codon usage (RSCU) of 64 codons as a measure of codon usage bias between groups, where the dashed line represents the threshold of values if >1.0, the codon is used more often than expected, and when <1.0, the codon is used less often than expected. **c** Density plots of protein isoelectric point (pI) (left) and purine-loading index (PLI) (right) distributions between groups. **d** Guanine–cytosine percentage (GC%) (left) of protein-coding sequences (CDSs) and protein lengths (right). **e** Predicted optimal growth temperature of five vMAGs (bins) and twelve reference genomes, where the x-axis corresponds to temperature (in Celsius). In each panel, asterisks represent significant statistical differences between the two virus groups (i.e., vMAGs versus reference viral genomes).

Putative giant Nucleocytoviricota virus homologs were found in the majority of vMAGs (n = 12) (Fig. 4c). To delineate families within this group, we searched for markers using GVOGs)[33] and NCVOGs[43] databases (Supplementary Data 6), together with homologs in databases, to build maximum likelihood (ML) phylogenies. Individual phylogenies for nine marker genes (Fig. 4a, Supplementary Fig. 1) confirmed the initial taxonomic identification for some vMAGs as Nucleocytoviricota and placed two unclassified vMAGs (bin787 and bin795) into the Pithoviridae and Asfarviridae families, respectively. We resolved their phylogenetic positions within these families and found that vMAGs affiliated with Mimiviridae-like families clustered with Tupanviruses isolated from the deep-sea[44], Pithoviridae-like vMAGs clustered with Pithoviruses also retrieved from deep-sea, in the Loki's Castle hydrothermal vent system[45], Pandoravirus-like clustered at the base of Pandoraviruses isolated from Siberian permafrost and from aquatic environments[46,47], and one Asfarviridae-like vMAG clustered at the base of the main group, near Faustoviruses[48] and Pacmanviruses[49] that are distantly related to Asfarviridae. Despite their inclusion in established giant virus families, vMAG average amino acid identities (AAI) indicate high divergence when compared to viral reference genomes (Supplementary Data 9). Given their relative phylogenetic positions, it appears that hot springs giant vMAGs split early on from most of their relatives. These findings, together with the vast geographic distribution of viruses within these viral families from deep-sea locations in South America, the North Sea, and the Siberian peninsula, are consistent with the ancient splits of vMAGs.

## Footprints of genome-wide adaptation to extremophily at Lemonade Creek

The genomes and proteomes of thermophilic bacterial and archaeal species differ from those in mesophilic species due to local adaptation[50–52]. We investigated whether the same holds for hot springs vMAGs (n = 25) by comparing their genomes to those of phylogenetically closely related viruses, not from Lemonade Creek (n = 46) (Supplementary Data 7). To achieve this aim, we compared amino acid frequencies, including the thermophilic amino acid signature (IVYWREL—Ile, Val, Tyr, Trp, Arg, Glu, and Leu)[50], relative synonymous codon usage (RSCU), protein isoelectric point (pI) and length, purine-loading index (PLI), guanine-cytosine percentage composition (GC%), and predicted optimal growth temperatures (pOGT).

We find that hot springs vMAGs have a marginally higher frequency of the amino acids Cys, Gln, Glu, His, Leu, Phe, Pro, Ser, The, Trp, and Tyr, and lower Ala, Asp, Gly, Ile, Lys, Met, and Val content, with similar frequencies of Arg when compared to viral genomes from other environments. However, none of these values are statistically significant (Fig. 5a). Comparison of the amino acids IVYWREL also showed no statistical difference (p = 0.3888), although there was a higher mean frequency in hot spring vMAGs (5.62%) than in non-hot springs virus genomes (5.42%). For RSCU (Fig. 5b), we found significant differences between groups, with AGG and ATA as preferred codons in hot spring vMAGs, whereas CGC, CTC, GGC, TAC, TCG, TGC, and TTT are preferred in non-hot spring viral genomes (p < 0.0005). We also found that hot spring vMAGs have a higher protein pI (p < 0.0005) (Fig. 5c left), higher PLI (p < 2.538e$^{-11}$) (Fig. 5c right), lower GC% (mean 38.09%; non-hot spring viruses 41.87%) (p < 2.2e$^{-16}$), and

shorter protein lengths ($p < 2.2e^{-16}$) (Fig. 5d). Lastly, we predicted optimal growth temperatures (pOGT) for the longest vMAGs ($n = 5$) and non-hot springs viruses ($n = 12$) (Fig. 5e) and found that mean pOGT for the former (55.3 °C) was significantly higher than for the latter (46.6 °C) (W = 52, $p = 0.01939$; Supplementary Data 7). These findings indicate that Lemonade Creek vMAGs have genomic signatures of thermophilic adaptation, suggesting a long-term relationship with hot spring environments.

## Discussion

A major finding of our study is that even though different habitats at hot springs may be adjacent (within centimeters of each other) and visually dominated by Cyanidiophyceae, multiple, divergent virus classes exist at these sites that are often habitat specific (Fig. 1c). Megaviricetes is the dominant class at Lemonade Creek, suggesting they may infect and influence red algal population dynamics. This idea is supported by the finding of vHGTs within this group[21,22,53]. Most virus proteins in our dataset are unclassified, indicating remarkable novelty (i.e., based on the current data) at geothermal sites, with some having roles in genome mobility (e.g., endonucleases, transposases).

When comparing virus genomes from Lemonade Creek to those of related lineages isolated from mesophilic sites, our results demonstrate the existence of genome and protein signatures associated with a thermophilic lifestyle. The early branching phylogenetic position of the most abundant Nucleocytoviricota vMAGs strongly supports Hypothesis 1 (H1) that we posed, whereby hot springs viruses are ancient lineages that have adapted to, and likely remained in geothermal environments, at least 1.5 Bya, since the time of Cyanidiophyceae origin[5]. These splits likely occurred when their hosts initially colonized extreme environments, emphasizing the ancient and divergent nature of hot springs vMAGs when compared to their virus relatives. Our work also expands to terrestrial thermal springs, the distribution of giant viruses that were previously identified in different extreme environments, such as the deep sea.

It has been proposed that the ancestral state of prokaryotes, and life itself, was thermophilic[54–56] because cells originated in the late Hadean or early Archean eon on the early Earth[57,58], potentially in a hot springs environment[59]. Recently, a major deep-diverging thermophilic bacterial clade from hydrothermal vents was reported[60]. However, a non-thermophilic origin was also suggested by placing the root of these lineages within Chloroflexota or Candidate Phyla Radiation (CPR). In another study, the root was placed between Gracilicutes and Terrabacteria, both of which contain thermophiles, but the majority are mesophilic[61–63]. Although we do not know if the ancestors of Lemonade Creek viruses were thermophilic, our results indicate that for at least 3 different viral groups (Nucleocytoviricota, CRESS DNA viruses, and Tectiliviruses), the shift to thermophilic environments was an early event in their evolution.

Thermophilic organisms evolved several features to enhance genome and protein thermostability, such that proteomes and single protein ortholog pairs can be differentiated from homologs in mesophilic species based on sequence and composition properties[64]. Specifically, the summed frequency of the Ile, Val, Tyr, Trp, Arg, Glu, and Leu amino acids are correlated with the optimal growth temperature (OGT) and thermal adaptation in Bacteria, Archaea, and Eukarya[50,65]. The average length of proteins is much shorter[66–68], and a higher protein isoelectric point (pI) of thermophilic Bacteria and Archaea reflects not only high temperature but also adaptation to acidic environments[69,70]. Codon bias in thermophiles is related to protein thermostability by enhancing the number of ionic interactions and salt bridges[71], reflected by the preference for AGR codons (AGG, AGA), that encode arginine (Arg), contrasting to CGN codons preferred by mesophiles[72]. A high purine-loading index (PLI - the difference between purines and pyrimidines) prevents the formation of double-stranded RNAs generated by RNA-RNA interactions that occur with increasing temperatures[73]. In prokaryotes, the OGT is related to the average PLI of open reading frames and is strongly correlated with codon usage bias[50], as in plastid genomes from the extremophilic red alga *Cyanidium caldarium*[74].

Our findings are consistent with previous studies on extremophilic cellular organisms and reveal a fundamental principle of biology at the extremes: hot spring viruses are not simply different with respect to minor features from relatives inhabiting mesophilic environments, rather, they have evolved distinct molecular signatures characteristic of their hosts, i.e., thermophilic Bacteria, Archaea, and possibly Eukarya. This suggests that adaptation at the extremes follows the same underlying rules, regardless of the domain, specifically, ancient origins, compact proteins, stable molecular bonds, optimal growth at higher temperatures, and potentially, communal interactions through infection or syntrophy. Our study provides the first assessment of viruses associated with polyextremophilic red algae that form extensive microbial mats in YNP and are common worldwide at geothermal sites[2,5,75]. Our study identified only four vOTUs that map to RNA viruses (Riboviria), likely due to a bias favoring DNA viral genome amplification rather than the absence of these viruses in hot springs[76–78]. Despite the challenges posed by extreme conditions in YNP, such as high heat and acidity that may compromise RNA stability, further exploration of RNA viruses in these mats is necessary to understand their ecology and potential genome adaptations under extreme conditions.

Finally, our metagenomic data demonstrate that viruses from red algal mats at Lemonade Creek are phylogenetically diverse, abundant, and appear to have persisted in extreme environments at least for as long as the Cyanidiophyceae have existed[79]. Although more recent than the earliest forms of prokaryotic life on our planet, these eukaryotic algal mats span a long period of the Earth's history, surviving and thriving in some of the most challenging ecosystems for both cellular life and viruses.

## Methods
### Site description and sampling
The geothermal study site in Yellowstone National Park (YNP) is referred to as Lemonade Creek (Lat.: 44° 48'05" N, Long.: 110° 43'44" W). During the spring months, it drains run-off from snow melt (March–May), and in the following months of the year, it is completely derived from four geothermal tributary features (Fig. 1a). Samples were taken downstream of the four tributaries to reflect the inputs from all sources. Three environments were sampled at this site: (1) the lush green biofilms formed mainly by *Cyanidioschyzon* in the creek channel (Fig. 1b), (2) the soils adjacent to the creek channel, and (3) the endolithic materials also adjacent to the creek channel. On the day of sampling, the creek water pH and temperature were 2.55 and 44 °C, respectively. The adjacent soil pH was 2.1, and the temperature was ~32 °C. Replicate ($n = 4$) samples were taken from each environment on the same day. For the creek biofilm samples, small stream bed rocks were loaded into 50 ml sterile Falcon tubes along with ~15 ml of creek water and then shaken vigorously (~10 s) to generate a thick suspension of biomass. Rocks and grit were allowed to settle (~10 s), and then the suspension was decanted into a 15 ml Falcon tube and immediately centrifuged ($2000 \times g$ for 3 min). The resulting pellet was saved, and the supernatant was discarded. Soil samples were taken by scraping the upper ~0.5 cm of surface soil, being careful to avoid soil below the discrete algal layer (Fig. 1b). Endolithic material was acquired by breaking off chips of stream-side rock. Immediately after the acquisition, the samples were flash-frozen in an ethanol-dry ice bath and kept on dry ice during transport back to the lab, where they were kept at −80 °C until nucleic acid extraction. The fieldwork and sample collection were conducted by Timothy McDermott under NPS research permit YELL-2022-SCI-5364.

### Nucleic acid extraction
Total DNA was extracted from the samples with the Qiagen DNeasy PowerSoil Pro kit (Hilden, Germany). Samples were checked for quality, purity, and concentration using the Nanodrop 3000C and Qubit 2.0. They were stored at −80 °C and shipped on dry ice to the DOE Joint Genome Institute (JGI) for sequencing. Due to the challenges associated with extracting biological material from the endolithic site, sample E4 underwent PCR amplification during library preparation to increase DNA concentration to a usable level.

## Sample sequencing and metagenome assembly

DNA samples were sequenced by the DOE Joint Genome Institute (JGI). For each sample, an Illumina library was constructed and sequenced 2 × 151 using the Illumina NovaSeq S4 platform following SOP 1064 (the number of reads and bases sequenced for each sample are listed in Supplementary Data 1). BBDuk (v38.94)[80] was used to remove contaminants, trim reads that contained adapter sequence, trim G homopolymers of size 5 or more at the ends of the reads, and "right quality" trim reads where quality drops to 0. BBDuk was used to remove reads that contained 4 or more 'N' bases, had an average quality score across the read less than 3, or had a minimum length ≤51 bp or 33% of the full read length. Reads mapped with BBMap[80] to masked human, cat, dog, and mouse references at 93% identity were removed from downstream analysis, as were reads that aligned to common microbial contaminants (following SOP 1077). The reads retained after filtering were corrected using bbcms (v38.90; "-Xmx100g mincount=2 highcountfraction=0.6") and assembled independently (i.e., the corrected reads from each library were assembled separately) using spades v3.15.2 ("-m 2000 --only-assembler -k 33,55,77,99,127 --meta")[81].

## Identification of viral scaffolds

Viral scaffolds were identified in each of the 12 assembled metagenomes by taking scaffolds longer than 1.5 kb and applying VirSorter2 ("--min-score 0.9 --min-length 1500 --hallmark-required-on-short")[82], geNomad v1.3.0 ("end-to-end --min-score 0.7 --cleanup --splits 8"; https://github.com/apcamargo/genomad)[83], and a custom pipeline using DIAMOND v2.0.15 BLASTx (e-value 1e-5, "--ultra-sensitive")[84] against the viral protein reference sequence database (v211) from GenBank and the IMG/VR v4 database[85]. The latter approach was used to retrieve shorter viral scaffolds with no hallmark genes that would normally be excluded from downstream analysis by VirSorter2 and geNomad. We combined the results of these tools and extracted scaffolds resulting from these searches; Prodigal v2.9.0 ("-p meta")[86] was used to predict proteins in the resulting scaffolds. To remove false positives, we aligned the predicted proteins with DIAMOND BLASTp against the non-redundant GenBank database (NR), with taxonomic information for each protein included in the output – specifically the highest taxonomic level classification (i.e., the "sskingdoms" field) for each hit. We combined this blast search with results from the previous blast analysis against the IMG/VR v4 database, keeping only the scaffolds where most protein top hits (≥60%) per scaffold were with viruses. Finally, we used CheckV v1.1[87] to flag contaminant scaffolds ("checkv end_to_end", default parameters) which were subsequently removed. The resulting scaffolds that passed CheckV filtering were designated as viral scaffolds; this final set of scaffolds had an average scaffold length of 3.97 kb, with 703 of the scaffolds derived from the creek samples, 2239 from the endolithic samples, and 3348 from the acidic soil samples.

## Taxonomic classification of virus scaffolds

Taxonomic classification of viral scaffolds was done using the parsed geNomad output and a majority rule approach using DIAMOND BLASTp top hits from the predicted proteins against the non-redundant Genbank (NR) and IMG/VR v4 databases, processed using a custom Python script (https://github.com/LFelipe-B/YNP_Lemonade_Creek_viruses).

## Clustering of virus scaffolds into virus operational taxonomic units (vOTUs)

Predicted viral scaffolds that shared >95% average nucleotide identity and >85% alignment fraction were clustered and dereplicated into vOTUs using BLASTn ("-outfmt '6 std qlen slen' -max_target_seqs 10000") and the anicalc and aniclust python scripts ("--min_ani 95 --min_tcov 85 --min_qcov 0") available with CheckV (https://bitbucket.org/berkeleylab/checkv/src/master/scripts/). Clustering thresholds were chosen according to community standards[88].

## vOTU abundance across samples

The abundance of the identified vOTUs was computed by CoverM v0.6.1 ("--min-read-percent-identity 98 --min-covered-fraction 0"; https://github.com/wwood/CoverM) using corrected metagenome reads aligned with BBMap v38.87 ("ambiguous=random")[80] against the vOTU sequences as well as contigs from other cellular organisms identified in the samples. The Reads Per Kilobase per Million mapped reads (RPKM) value was extracted for each metagenome sample and used for downstream analysis. We chose to perform downstream abundance analysis at the "class" taxonomic rank as this is the most complete rank across all viral lineages. That is, some viral groups, such as Caudoviricetes, lack an updated lower taxonomic classification, making its analysis and comparison with other lineages with, for example, family-level classifications, challenging. Taxonomic annotations where no "class" level information is described (i.e., environmental samples, unclassified archaeal viruses, unclassified viruses, or where only the realm was present) are considered "Unclassified" in our analysis.

## vOTU alpha and beta diversity community analysis

Alpha diversity was estimated by calculating richness (Chao1 index), diversity (Shannon), and evenness with vegans at the class taxonomic level. To test the effect of sample type on the virus composition, we run a Permanova analysis (999 permutations) on a Bray–Curtis dissimilarity matrix constructed from the viral classes using the *adonis2* function from vegan[89] R package. To assess the sample clustering profiles, we then use the same dissimilarity matrix in a non-metric multidimensional scaling (NMDS) analysis using Vegan, with the dimensions plotted using the ggplot2 R package. We used the Shapiro test to assess the normality of the data. When the data were not normally distributed, we ran the non-parametric Kruskal-Wallis test, followed by the post-hoc Tukey–Kramer–Nemenyi test with p-values adjusted using Bonferroni correction.

## Functional prediction of vOTU proteins

We used eggNOG-mapper v2.1.6 ("--pfam_realign denovo --report_orthologs")[90,91] to functionally annotate the proteins predicted in the vOTUs. Gene ontology terms (GO terms; http://geneontology.org/docs/download-ontology/) were used to evaluate the functional landscape of the vOTUs, discarding the "obsolete" and generic terms before analysis: biological process (GO:0008150), biosynthetic process (GO:0009058), cellular process (GO:0009987), cellular component (GO:0005575), molecular function (GO:0003674), metabolic process (GO:0008152). We also use the KEGG (Kyoto Encyclopedia of Genes and Genomes — https://www.genome.jp/kegg/kegg1.html and PFAM (Protein families; https://www.ebi.ac.uk/interpro/entry/pfam/) databases to evaluate biological pathways and protein domains for vOTU predicted proteins (Supplementary Data 4).

## Sequence similarity network construction with unclassified vOTU proteins

Viral protein clusters (vPCs) were derived from sequence similarity networks constructed using unclassified vOTU proteins combined with their top hits against GenBank NR and IMG/VR v4 databases. Before clustering, CD-HIT v4.8.1[92], run using a 60% identity and 80% coverage cutoff (-c 0.60 -s 0.80 -n 4), was used to cluster similar proteins. The representative sequence from the CD-HIT analysis was then compared together using an all-against-all DIAMOND BLASTp (e-value 1e-5, "--ultra-sensitive") search, with best reciprocal blast hits removed using a custom python script (available at https://github.com/Lfelipe-B/YNP_Lemonade_Creek_viruses). The results were constructed into a network by the igraph v4.2.1R package[93], with singleton proteins and less connected components with degrees < 8, removed from the final network before visualization (Supplementary Data 4).

## vOTU horizontal gene transfer analysis

To search for horizontal gene transfer (HGT) candidates between cellular life and viruses, we calculated the Alien index (AI)[94] and the HGT index

(hU)[95] for each protein in the vOTUs, grouping the results by viral class. We parse the full taxonomic annotations of the subject accession numbers from the DIAMOND BLASTp outputs against the non-redundant GenBank database (NR) using custom Python scripts (available at https://github.com/LFelipe-B/YNP_Lemonade_Creek_viruses). Hits were categorized as "non-alien" if the subject sequence was from a virus protein and as "alien" if the subject sequence was from a eukaryote, bacteria, or archaea; hits to the same viral phylogenetic group of the vOTU sequences were omitted from the subsequent calculations to avoid self-hits with sequences from the same or similar taxa already submitted to GenBank. vOTU proteins that had AI and hU indices ≥30, indicating that they have better scoring hits to non-viral proteins than to viral proteins, were considered putative candidates for HGT between viruses and cellular taxa. The putative sources of HGTs in each viral class was evaluated using the domain and/or phylum taxonomic rank of the best cellular hits from each putative HGT (Supplementary Data 5). Available scripts for this analysis at https://github.com/LFelipe-B/YNP_Lemonade_Creek_viruses.

### Virus scaffold binning into virus metagenome-assembled genomes (vMAGs)

The corrected reads from each of the metagenome samples were aligned against each of the metagenome assemblies using BBMap v38.87[80] ("ambiguous=random rgid=filename"), with the resulting mapped reads sorted using samtools sort[96] v1.11. For each metagenome assembly, the "jgi_summarize_bam_contig_depths" tool from MetaBAT2[97] v2.15 was used to calculate the scaffold read mapping depth in each of the corresponding bam files produced from the mapping of the corrected metagenomic reads. The depth information for each of the vOTUs was extracted from their respective metagenome assemblies and used by Metabat2 to produce viral bins, which were assessed for quality and completeness using CheckV. Bins that were considered with completeness as low (30–50%), medium (50–90%), high (>90%), or complete (100%) according to CheckV quality score were retained for downstream analysis and subsequently referred to as viral metagenome-assembled genomes (vMAGs). There were, in total, 1082 vOTUs recruited into the vMAGs. Taxonomic annotation as performed by a majority consensus rule using top hits from all the predicted ORFs in each vMAG, with the lowest taxonomic rank that still satisfies the majority threshold (60%) assigned as a vMAG's final classification.

### vMAG viral marker genes and phylogenetic profile

We screened vMAGs for the presence of viral marker genes to delineate lower taxonomic ranks (below class level) and solve their specific phylogenetic relationships. For this we checked the predicted annotations, DIAMOND BLASTp output files from previous initial searches, and for the presence of giant virus hallmark genes from the Nucleo-Cytoplasmic Virus Orthologous Groups (NCVOGs)[43] using DIAMOND BLASTp, and for Giant Virus Orthologous Groups (GVOGs) using the tool "ncldv_markersearch" (https://github.com/faylward/ncldv_markersearch)[98]. Final taxonomic annotations, environmental and host sources of viral hits were retrieved from Giant Virus Database (https://faylward.github.io/GVDB/), IMG/V (https://img.jgi.doe.gov) and Virus Host database (https://www.genome.jp/virushostdb). To confirm the presence of NCVOGs and GVOGs in specific vMAGs we subjected candidate marker proteins from vMAGs to a BLASTp search using the online suite (https://blast.ncbi.nlm.nih.gov/Blast.cgi), removing false positives (i.e., wrong marker gene or non-viral sequence). We keep only nine markers that overlapped in both NCVOGs and GVOGs databases since these are the most well-conserved genes for reconstructing evolutionary relationships within this viral group[99]: B DNA Polymerase (PolB), A32-like packaging ATPase (A32), virus late transcription factor 3 (VLTF3), superfamily II helicase (SFII), major capsid protein (MCP), the large and small RNA polymerase subunits (RNAPL and RNAPS, respectively), TFIIB transcriptional factor (TFIIB), and Topoisomerase family II (TopoII) (Supplementary Data 6, Supplementary Data 8). Next, we combined proteins from GenBank and IMGV/V4 with vMAG candidates and performed a multiple sequence alignment (MSA) using MAFFT v7.305b (L-INS-i algorithm: "--localpair --maxiterate 1000")[100], with the resulting alignments trimmed using TrimAI v1.2 in automated mode (-automated1)[101]. Individual maximum likelihood phylogenies were built using IQ-TREE v2.0.3[102] with the ModelFinder option (-m TEST)[103] and 1000 bootstrap replicates (UFBoot)[104]. Phylogenies were midpoint rooted, visualized, and edited using the R packages ape v5.6-2[105], phangorn v2.10.0[106], ggtree v3.4.4[107], phytools v1.2-0[108] (Revell 2012), and ggnewscale v0.4.9 (https://github.com/eliocamp/ggnewscale) in the R environment (version 4.2.1). Average amino acid identity (AAI) when compared to reference viral genomes was calculated using the tool "AAI calculator from http://enve-omics.ce.gatech.edu/aai/" (Supplementary Data 9).

### Virus genomic signature analysis

We downloaded complete viral reference genomes from GenBank (https://www.ncbi.nlm.nih.gov/nuccore) for species that showed proximity to our viral proteins in individual phylogenetic marker gene trees or that had top hits against our vMAGs (full list and accession numbers in Supplementary Data 7). Amino acid frequency and protein isoelectric point (pI) were calculated for each protein per genome using the functions "AAstat" and "computePI" from the R v4.2.1 package seqinr v4.2-30[109]. To calculate relative synonymous codon usage (RSCU) tables from predicted coding sequences (CDS) we use the uco function ('index = "rscu"') from the seqinr package. The purine loading index (PLI) per CDS was calculated using the formula from Cristillo et al.[108] and Lambros et al.[73], where we first counted the proportion of purines $(A + G)$ and pyrimidines $(C + T)$, we then calculated $\Delta W = (A - T)/N * 1000$ and $\Delta S = (G - C)/N * 1000$, where $N$ is the total number of bases in the CDS and 1000 is to convert it to per kilobase, and finally the PLI $= \Delta W + \Delta S$. To calculate cytosine and guanine percentage (GC%) from each CDSs we used a custom Python script available at https://github.com/LFelipe-B/YNP_Lemonade_Creek_viruses. For the prediction of optimal growth temperatures for vMAGs and viral reference genomes, we use the tool "CnnPOGTP"[109] available at www.orgene.net/CnnPOGTP, which is based on deep learning of $k$-mers distribution from genomic sequence.

### Statistics and reproducibility

To test for differences in amino acid representation and codon usage between vMAGs and viral reference genomes, we used ANOVA, followed by Tukey's post hoc test. To compare genomic signatures among vMAGs and reference viral genomes, we used $t$-test or ANOVA, Wilcoxon rank test, or Kruskal–Wallis rank test in the case data was not normally distributed according to the Shapiro–Wilk normality test, with all $p$-values adjusted for multiple testing. Statistical analysis was conducted in R v4.2.1. The reproducibility of experiments is not relevant here. However, details about sample sizes and number of replicates and how replicates were defined are described above in the Methods section.

### Reporting summary

Further information on research design is available in the Nature Portfolio Reporting Summary linked to this article.

## Data availability

Files containing vOTUs and vMAGs fasta sequences, and vMAG newick phylogenetic trees are available at https://doi.org/10.5281/zenodo.10465700.

## Code availability

Scripts to run parts of this analysis are available at https://github.com/LFelipe-B/YNP_Lemonade_Creek_viruses.

**Article**

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

## Acknowledgements
L.F.B. was supported by the NASA Postdoctoral Program at Rutgers University, New Brunswick, administered by Oak Ridge Associated Universities under contract with NASA. J.V.E. was supported by National Aeronautics and Space Administration Future Investigators in NASA Earth and Space Science and Technology (FINESST grant 80NSSC19K1542). T.G.S. and D.B. were supported by a grant from NASA (80NSSC19K0462) awarded to D.B. D.B. was also supported by a NIFA-USDA Hatch grant (NJ01180). T.R.M. was supported by a grant from NASA (80NSSC21K0487). We would also like to acknowledge our collaboration with the JGI. The work (proposal: 10.46936/10.25585/60000481) conducted by the U.S. Department of Energy Joint Genome Institute (https://ror.org/04xm1d337), a DOE Office of Science User Facility, is supported by the Office of Science of the U.S. Department of Energy operated under Contract No. DE-AC02-05CH11231. Sampling at Lemonade Creek, YNP was conducted under Research Permit YELL-2020-SCI-5364 issued to T.R.M.

## Author contributions
D.B., L.F.B., and T.R.M. conceived and designed the project, sampling, and scope. K.B. and I.V.G supervised data collection at the Joint Genome Institute. T.R.M. and W.C.C. performed sampling at Yellowstone National Park. L.F.B, T.G.S, J.V.E., and T.J. analyzed data. L.F.B, T.G.S., J.V.E., T.R.M., and D.B. wrote the paper, and all authors reviewed and approved the final version of this paper.

## Competing interests
The authors declare no competing interests.
