## [Peer Review File · Communications Biology]

Reviewers' comments:

Reviewer #1 (Remarks to the Author):

In this study Benites and colleagues examine viral diversity in Yellowstone hot springs and nearby environments. The springs themselves are dominated by red algae and have other eukaryotic community members. The authors find a rich diversity of Megaviricetes (giant viruses) and Caudoviricetes (phages) and perform a suite of comparative genomic analyses to examine their evolution and ecology. They conclude that the viruses are likely endemic to the hot springs and may represent ancient lineages.

Overall, this is an exciting study that examines the viral diversity in a fascinating environment. The authors have a clear hypothesis-driven framework for how viral diversity is structured in these springs, which elevates the manuscript above a simple viral diversity catalog. Given the broad interest in extreme environments, this work will reach a wide range of readers. I have a few suggestions for some possible revisions that the authors may wish to consider that may clarify or strengthen some aspects of the study.

-The first two sentences of the abstract could be re-worded for clarity – right now it sounds like all hot springs are dominated by red algae, at least to me. Perhaps something like “Red algae of the class Cyanidiophyceae dominate microbial biomass of many hot springs, but little is known about host-virus interactions in these environments” or something like that?

-The pandoravirus-like MAG is fascinating. No pandoravirus has ever been described from a metagenome, so this is particularly novel and interesting. In the tree in Figure 4 it looks like it is closer to pandoraviruses than even mollivirus. Could the authors report the AAI, or maybe just the amino acid identify of PolB or some other markers, to cultivated Pandoraviruses for comparison (most of these form a fairly compact clade, so just using Pandoravirus dulcis, for example, should be fine). It would be nice to know just how similar it is to true Pandoraviruses. One could imagine that this is a valuable MAG for studying how gigantism evolved in this lineage, since it is a basal-branching pandoravirus.

-The authors must excuse my preoccupation with viral taxonomy, but this is something that I think about a lot and I believe a few clarifications here would strengthen the manuscript. The taxonomy of giant viruses has been in flux for the last few years, so I understand it is quite challenging to define the different groups, but a hierarchical taxonomy has been proposed for the Nucleocytoviricota (<https://doi.org/10.1371/journal.pbio.3001430>). Could the authors expand a bit on if these taxonomic levels were annotated in Figure 4? For example, it looks like the Mimivirus-like MAGs are clustered in between two different lineages of Mimiviridae, but I suspect that these may be a broader group of Imitervirales (previously most Imitervirales were considered to be Mimiviridae, but now this is just one family within the broader order). If the MAGs fall within the Mimiviridae it would be a bit counter to the idea that the hot spring viruses are early branching, but perhaps by annotating families according to a recent hierarchical taxonomy it would be easier to show that the hot spring MAGs are early branching with respect to families within the Imitervirales. In my group we developed an automated classification pipeline that may be useful for doing this (<https://doi.org/10.1101/2023.11.10.566645>)

-Line 64 – The Nelson et al study here actually used methods for detecting endogenous viral elements in eukaryotes that have an extremely high false positive rate of detection. So I think the last part of this sentence may come across as slightly misleading, because many of the HGT events described in this study are not from viruses. Please consider citing alternative studies of virus-to-eukaryote gene transfer or re-wording.

-Do the authors have any hypothesis about the hosts of the giant viruses? A giant virus of red algae

has not been discovered, so this could potentially be quite interesting, especially if it is a Pithovirus! Even if it is not conclusive, any speculation about the hosts of these viruses could be interesting, perhaps based on knowledge of other eukaryotic diversity that is present in the hot springs. I believe all pithoviruses enter their hosts through phagocytosis – are any red algae known to do this? Some recent work has suggested that some green algae can engulf bacterial prey (<https://doi.org/10.1038/s41396-021-00899-w>), but I wasn't sure about red algae.

-Did any viral proteins have homology and high percent identity to known red algal proteins? That may be one possible way to make a host prediction.

-Figure 3C. There have been widespread reports of HGT between giant viruses and bacteria (for example see Boyer et al <https://doi.org/10.1073/pnas.09113541>), but I always thought that some of this may be due to biases in the genome collection (i.e. lots of bacteria to recruit weak blastp hits, but relatively fewer eukaryotes). Would this bias in the genome collection impact these results? If not, it is quite interesting to note the Proteobacteria-giant virus connections.

-Line 92 – I think the 22 Mbp is a typo – it should be 2.2 Mbp, correct?

-For the unknown DNA virus – bin871 – I am curious if a broad classification at the Realm or Phylum level may be possible. It is a large DNA virus, which suggests it is either a member of the class Caudoviricetes (jumbo phage), phylum Mirusviricota, or phylum Nucleocytoviricota. The first two have an HK97 capsid and the last has a double-jellyroll capsid. These classifications are largely based on the capsid, so perhaps remote homology to a capsid could be detected? If there are some proteins with no annotation that are about 500 aa long, you could try an hhpred search, which I have found to be quite sensitive and may find a hit when other methods fail.

-Line 484 – unfortunately IMG/VR does not include an up-to-date database of giant virus genomes. A more complete set that has been decontaminated (i.e. cellular contamination removed) can be found in the GVDB, which also includes a hierarchical taxonomy for the viruses (<https://faylward.github.io/GVDB/>). I don't think this will impact the results here that much, so I don't think that remaking the trees is necessary, but the authors may wish to keep this in mind for future viral studies.

Congratulations on the exciting study.
Frank Aylward

Reviewer #2 (Remarks to the Author):

In the manuscript, the authors investigated the viral communities from samples of three distinct features below the tributaries of Lemonade Creek in Yellowstone National Park. This broadly entailed assessment of i) the taxonomic composition of the communities, ii) the likely role of the viral communities in genetic exchange among the different life-forms in these communities, and iii) the potential thermal adaptation of viral genomes in these communities. Overall, this work represents an interesting exploratory effort at a well-restricted, local scale, with unexpected results and rigorous analysis of data. The figures are also very well constructed and of high quality, although some minor suggestions on these are outlined below. Overall, I believe the manuscript would be of interest to a broader community, ranging from biologists working in extreme environments to evolutionary biologists, and shows pretty clearly that viruses should be analyzed with cellular community members, and that large differences in community structure can be found at very localized scales. The manuscript represents great, novel, and exciting science, and my suggestions are mainly associated with potential improvements that the authors may want to consider to the manuscript itself and its structure.

Some larger comments:

1. Overall, the manuscript did not read as comfortably as it could. The flow of the manuscript felt slightly disjointed, with some large-scale conclusions mentioned in the results in some sections (e.g., line 151-152 "These results suggest that hot springs harbor many novel proteins that may play specific ecological roles in each environment."), while at other times, the context is only provided in the discussion, with no context for the results being discussed in the results section (e.g., section on footprints of extremophily adaptations). These slight differences in writing style read as different contributors having written different sections of the manuscript and overall contribute to a slightly disjointed feel for the reader. I think a potential solution for this would be to combine the results and discussion for the manuscript, as providing the context and implications of the results as they are being discussed may be easier than deconvoluting sections that already contain some discussion in the results.

2. Furthermore, there are several areas later in the manuscript where the authors refer to analyses to address a particular aim (e.g., line 157 "For this aim,..."), however, these aims weren't clearly defined in the manuscript. Overall, I believe the reader would benefit from having the last paragraph of the introduction refined to have a clearer layout of the aims for the study. This would then also provide a clearer linkage to why certain aspects of the study was undertaken in a particular way. As I outlined above, my take home aims of the manuscript were "i) the taxonomic composition of the communities, ii) the likely role of the viral communities in genetic exchange among the different life-forms in these communities, and iii) the potential thermal adaptation of viral genomes in these communities". As such, if the manuscript was slightly more focused along these three aims explicitly, it would feel much more cohesive and easier to follow for the reader.

3. Although the authors have provided a lot of useful supplemental information, something that is slightly difficult to assess from the phylogenies provided is the clear link between early diverging lineages in certain viral classes and the rest of the members of those viral classes, and the linkage to thermal/extreme environments. This is because all of the phylogenies are collapsed at various nodes and one cannot really dig into the data to assess the relationships within those groups (e.g., whether obtained relationships reflect previously obtained groupings thus supporting the robustness of the trees/ whether any of the novel taxa fall within collapsed branches etc.). A potential solution for this would be to also include either uncollapsed phylogenies as PDFs in the supplement, or to include the newick files of the actual trees as supplements. Furthermore, something that I would find useful would be mapping environments or thermal or acidic conditions onto the trees for the viruses included in the analyses, or minimally to Figure 4. I think this would give a greater visual representation of those early diverging viral lineages being mostly extreme environment associated and would give an easier, visual reach to the early Earth environments discussions.

4. Additionally, with all of the data presented, I do find the discussion of the significance of the various aspects slightly short, and more analogous to a conclusion section. I realize there are word limits that the authors have to adhere to, but perhaps consider having supplemental text if appropriate to expand on the discussion of various aspects? This would then also afford the authors space to really tailor the results and discussion to hitting those aims hard, and move sections that detract a little from the main story to the supplement (e.g., most information in the section on the vMAGs etc.).

Minor comments:

Line 46: "in the modern Earth" suggest to change to "on the modern Earth" unless the authors are specifically referring to subsurface extremes.

Line 92: I think this must be a typo for the largest vMAG being 22 Mb? Elsewhere it is indicated as 2.2 Mb, and this is ~10x larger than most prokaryotic thermophile genomes, more on par with some eukaryotic genomes.

Line 107-109: Combined with the comment for Fig. 1 (see below), you could change this sentence then to "Although *Maveriviricites* only comprise 0.5% of the total vOTU count, they are present in all environments (*referring to a new supplemental table for specifics*), which is expected.."

Line 116-120: A little expansion on why the endolythic samples might be so varied compared to the soil or creek samples would help contextualizing the result. Is this a common trend? Are there specific factors influencing the endolythic samples (e.g., distance from spring/rate of dessication etc.) compared to other samples? etc.

"Functional profile of the Lemonade Creek virus community" section - the authors state here that the viruses could participate as potential active members in various geochemical cycles, but I am missing clear links that support this claim in text. It is difficult for readers to contextualize these conclusions without clear evidence discussed in text, e.g., specific genes associated with the different cycles being discussed (if not in the main text, at least in supplemental text?).

Line 195-196 and 203-204: Some abbreviations are not expanded on first use, but only upon subsequent uses, e.g., GVOGs and NCVOGs - please check this carefully.

For figures:

Overall, the font size of most figures may make it slightly difficult to read the content, thus an overall increase in font size for all the multi-panel figures would be great if possible.

Fig. 1: Perhaps combining all portions of the population that occur in <1% into an "other" category as opposed to listing them individually, as the vast majority of the taxa are not visible in the bar graph, and then including a supplemental table with the actual full proportions/percentages? This would make the panel feel much less cluttered, and clearly highlight those differences among the highly abundant taxa among the samples.

Fig. 3c: Showing which domains of cellular life is referred to with the different phyla etc. would greatly help. Thus, ordering the key into discrete cellular domain groups and then adding brackets for indicating where those domains are on the upper axis of the figure would assist in making the figure clearer.

Fig. 4a: Having environmental context mapped onto this phylogeny would help a ton with visually showing the linkage between the earlier diverging lineages and extreme environments.

Fig. 5 I think expanding panels A and B to stretch the full width of the page, and then having panels C, D, and E below A and B would greatly increase the readability and layout of the figure.

Reviewer #3 (Remarks to the Author):

In this study the authors explore viruses attuned to geothermal habitats, specifically at Lemonade Creek in Yellowstone National Park. They assessed function (GO annotations and other methods) and acquisition of genes (i.e., HGT) using metagenomic sequencing to search for virus adaptations to these environments where their extremophilic hosts persist. Sampling efforts focused on three environments: (1) creek biofilms produced by Cyanidioschyzon (red alga), adjacent (2) soils and (3) endolithic materials. Their efforts uncover signatures of a "thermophilic lifestyle", and authors conclude that the virome at these geothermal habitats present ancient adaptations that have been maintained alongside the red algae found at the site (1.5 Bya). Broadly, the authors conclude that their viromes present evidence of the same "underlying rules" of adaptation to these environments that occur among other domains—with practical mention of the physical limitations of proteins (i.e., limited length) and managing folding (i.e., thermostability and the prevention of RNA-RNA interactions) with context to codon bias and other signatures within their data. To meet their conclusions the authors use several well supported programmes for their bioinformatics pipeline/workflow, including the use of viral markers to achieve stringent results. I want to call specific attention to the authors' dedication to making public their custom scripts—thereby providing transparency and reproducibility of the manuscript and broadly contributing to the community which likely includes others wanting to make use of their workflow. I enjoyed the manuscript and my comments are primarily out of interest/curiosity.

Additionally, this manuscript represents novel findings and identifies viruses of interest from several (putative) groups in an environment largely unexplored. It also makes larger conclusions about the domain of viruses overall in reference to the specific environment—which will be of interest to the

greater virus community.

COMMENTS

(1) Was anything done to enhance the retrieval of viral sequences from the source material, for example DNase application to rid "contaminant DNA" (i.e., nucleic acids outside of viral capsules/non-viral origin DNA). I do not see mention of this in the extraction portion of the methods, however I know JGI in-house processing can include additional steps. Mostly, I am curious about the potential (some) of your identified viruses to be (anciently + degraded) endogenous viruses given the results of Figure 4C where many bins are missing an identifiable MCP (which could be lost or made unidentifiable over time). Given that presence of an MCP gives substantial evidence of a virion stage, and that no or minimal efforts to remove "contaminant" eukaryote or prokaryote DNA could result in sequencing viral inserts from non-viral genomes I am wondering how likely you think this could be the case for some of these bins? Your methods do a great job of removing "contaminants" so I am wondering if viral inserts could be leftover from eukaryotic or prokaryotic genomes.

(2) Additionally, I am interested in bin750 given the large number of MCP hits it contains, with the current legend I cannot tell the estimate number of MCPs that were identified (seems to be >8 at least). Although I understand the paper is focused on adaptations to this extreme environment, have you done any work to evaluate potential hosts of these viruses, if bin750 is truly coding for so many MCP it could be reflective of its ability to infect your most abundant eukaryote(s) in the system. Given this, I am also interested in which samples the majority of these sequencing reads bin750 originated from (of the three environments). However, I understand this may be outside the scope of your study. I do feel that broadly the sample origin of the VMAGs (i.e., reads mapping reports) is an important component of the story that is missing given that the three environments are seemingly different (enough)—hopefully I have not simply missed this information somewhere.

(3) Going back to the phylogenetic analyses: ModelFinder test was used within IQTREE, which model(s) were established for the trees? Can you also report the size of the alignment that each tree is based off of? I am asking given the use of --automated in trimAl, which can be quite aggressive on virus data.

(4) How was sample E1 processed with the biodiversity indices, given that it was amplified? I can see by Figure 1C it is likely contributing to the variation shown on your indices. Do you have some ideas of why E1 is also so different or else why E2 and E3 happen to be so similar compared to the other endolithic samples? Furthermore, how were the 51% of unclassified vOTUs handled within the biodiversity indices given that no class level was achieved? Were these excluded from the analyses or have I misunderstood? More specifically, in Figure 1 are the "unclassified" in 1B represented in the indices of 1C? If so then how? If not then do you think they could contribute in any meaningful way to your results?

(5) What are these Circoviridae? CRESS viruses as a group can infect microbes, however Circoviridae specifically are typically associated with metazoans, I am surprised you recovered them with high quality + high completeness. What is your impression of these classifications and their potential hosts in this environment?

(6) Finally, I think the manuscript needs a bit of context within the complete virosphere. The paper features ss and dsDNA viruses, discusses prokaryotic and eukaryotic viruses, however there is no mention at all about the existence (or potential existence) of RNA viruses in geothermal habitats. Given that the larger context of the paper is a comment of viruses functioning similar to other domains in terms of adaptation to these environments it would be worth it to cover any studies related to RNA viruses at these sources, or a mention of the lack-there-of. Especially given that the biofilms are focused on a eukaryote (red alga), and RNA viruses can play important roles in eukaryotes-or at least infect them. For example, the preprint paper "Distinct groups of RNA viruses associated with

thermoacidophilic bacteria" (Urayama et al.; doi.org/10.1101/2023.07.02.547447) is exploring this (albeit in prokaryotes). I say this recognising that capturing RNA viruses requires preplanning, however I think they are a component missing from the paper overall given its larger commentary.

To the authors: thank you for sharing your research.
Sincerely,
E.E. Chase

Reviewers' comments:

Reviewer #1 (Remarks to the Author):

In this study Benites and colleagues examine viral diversity in Yellowstone hot springs and nearby environments. The springs themselves are dominated by red algae and have other eukaryotic community members. The authors find a rich diversity of Megaviricetes (giant viruses) and Caudoviricetes (phages) and perform a suite of comparative genomic analyses to examine their evolution and ecology. They conclude that the viruses are likely endemic to the hot springs and may represent ancient lineages.

Overall, this is an exciting study that examines the viral diversity in a fascinating environment. The authors have a clear hypothesis-driven framework for how viral diversity is structured in these springs, which elevates the manuscript above a simple viral diversity catalog. Given the broad interest in extreme environments, this work will reach a wide range of readers. I have a few suggestions for some possible revisions that the authors may wish to consider that may clarify or strengthen some aspects of the study.

-The first two sentences of the abstract could be re-worded for clarity – right now it sounds like all hot springs are dominated by red algae, at least to me. Perhaps something like “Red algae of the class Cyanidiophyceae dominate microbial biomass of many hot springs, but little is known about host-virus interactions in these environments” or something like that?

We thank reviewer 1 for the positive comments and support of our work. We have incorporated the suggested modifications to the abstract as follows: “Geothermal springs house unicellular red algae in the class Cyanidiophyceae that dominate the microbial biomass at these sites. Little is known about host-virus interactions in these environments”.

-The pandoravirus-like MAG is fascinating. No pandoravirus has ever been described from a metagenome, so this is particularly novel and interesting. In the tree in Figure 4 it looks like it is closer to pandoraviruses than even mollivirus. Could the authors report the AAI, or maybe just the amino acid identify of PoIB or some other markers, to cultivated Pandoraviruses for comparison (most of these form a fairly compact clade, so just using Pandoravirus dulcis, for example, should be fine). It would be nice to know just how similar it is to true Pandoraviruses. One could imagine that this is a valuable MAG for studying how gigantism evolved in this lineage, since it is a basal-branching pandoravirus.

We now include AAI values (Supplementary table 9) for all the largest vMAGs with medium and high completeness (including the Pandoravirus-like bin800), using reference viral genomes that clustered with our vMAGs in marker gene phylogenies. We also included a brief discussion of this new analysis in P8 L218. The identified Pandoravirus vMAG has a higher AAI (39.01%) with *P. dulcis*, than with *Mollivirus* spp., and with other Pandoraviruses.

-The authors must excuse my preoccupation with viral taxonomy, but this is something that I think about a lot and I believe a few clarifications here would strengthen the manuscript. The taxonomy of giant viruses has been in flux for the last few years, so I understand it is quite challenging to define the different groups, but a hierarchical taxonomy has been proposed for the Nucleocytoviricota (<https://doi.org/10.1371/journal.pbio.3001430>). Could the authors expand a bit on if these taxonomic levels were annotated in Figure 4? For example, it looks like the Mimivirus-like MAGs are clustered in between two different lineages of Mimiviridae, but I suspect that these may be a broader group of Imitervirales (previously most Imitervirales were considered to be Mimiviridae, but now this is just one family within the broader order). If the MAGs fall within the Mimiviridae it would be a bit counter to the idea that the hot spring viruses are early branching, but perhaps by annotating families according to a recent hierarchical taxonomy it would be easier to show that the hot spring MAGs are early branching with respect to families within the Imitervirales. In my group we developed an automated classification pipeline that may be useful for doing this (<https://doi.org/10.1101/2023.11.10.566645>)

Thank you for these comments. In response, we have revised Figure 4a and Extended Data Fig. 1 file with updated taxonomies for these viral groups. We have annotated the taxonomic levels in figures and supplementary materials, while also indicating in the Methods section the sources such as the “Giant Virus Database” (<https://faylward.github.io/GVDB/>). Whereas some vMAGs fall inside Mimiviridae, we do not claim they were the earliest divergence within this viral group, however they do have an early branching position, for instance with respect to the major subfamily, Mimivirinae.

-Line 61 – The Nelson et al study here actually used methods for detecting endogenous viral elements in eukaryotes that have an extremely high false positive rate of detection. So I think the last part of this sentence may come across as slightly misleading, because many of the HGT events described in this study are not from viruses. Please consider citing alternative studies of virus-to-eukaryote gene transfer or re-wording.

We concur with this comment, however to the best of our knowledge the Nelson et al. study is only one of two studies that reported viral HGTs with Cyanidiophyceae red algae, therefore we think it is reasonable to cite this study.

-Do the authors have any hypothesis about the hosts of the giant viruses? A giant virus of red algae has not been discovered, so this could potentially be quite interesting, especially if it is a Pithovirus! Even if it is not conclusive, any speculation about the hosts of these viruses could be interesting, perhaps based on knowledge of other eukaryotic diversity that is present in the hot springs. I believe all pithoviruses enter their hosts through phagocytosis – are any red algae known to do this? Some recent work has suggested that some green algae can engulf bacterial prey (<https://doi.org/10.1038/s41396-021-00899-w>), but I wasn't sure about red algae.

-Did any viral proteins have homology and high percent identity to known red algal proteins? That may be one possible way to make a host prediction.

Because we only had preliminary data regarding the cellular composition of these metagenomes at the time of our study, we could not perform any host-virus association analyses, such as co-occurrence, although HGT analysis could indicate past associations (discussed in P6 L157). Therefore, because we lack stronger evidence that could indicate specific linkages between viruses and hosts, we avoided speculation on this topic, leaving it as an avenue of future research. To the best of our knowledge, there is no report of red algae being able to perform bacterial phagocytosis. Even though we had reported strong HGT index hits with the red alga *Galdieria* (Cyanidiophyceae) (see in Supplementary table 5), however we found many more hits against other cellular organisms (such as amoebae and bacteria).

-Figure 3C. There have been widespread reports of HGT between giant viruses and bacteria (for example see Boyer et al <https://doi.org/10.1073/pnas.09113541>), but I always thought that some of this may be due to biases in the genome collection (i.e. lots of bacteria to recruit weak blastp hits, but relatively fewer eukaryotes). Would this bias in the genome collection impact these results? If not, it is quite interesting to note the Proteobacteria-giant virus connections.

Indeed, we are aware that it seems to be common to find bacterial hits to sequences in giant virus genomes. Because we used an enriched collection (IMG/VR) of environmental virus genomes, we think that this could alleviate the bias, but nonetheless, we still recover a higher amount of cellular hits with bacteria than with eukaryotes (or Archaea). One possible explanation for these results may be horizontal gene capture from the environment rather than true giant virus-bacterial associations in extant giant viruses.

-Line 91 – I think the 22 Mbp is a typo – it should be 2.2 Mbp, correct?

Thank you, we corrected it to 2.2 Mb.

-For the unknown DNA virus – bin871 – I am curious if a broad classification at the Realm or Phylum level may be possible. It is a large DNA virus, which suggests it is either a member of the class Caudoviricetes (jumbo phage), phylum Mirusviricota, or phylum Nucleocytoviricota. The first two have an HK97 capsid and the last has a double-jellyroll capsid. These classifications are largely based on the capsid, so perhaps remote homology to a capsid could be detected? If there are some proteins with no annotation that are about 500 aa long, you could try an hhpred search, which I have found to be quite sensitive and may find a hit when other methods fail.

We tried to find giant viral markers in bin871 using the tool “NCLDV markersearch (https://github.com/faylward/ncldv_markersearch) reported in the methods section P17 L481, and we also searched for sequence hits provided by the Mirusviricota paper. We did not find any hits with these datasets and tools. We did find hits against the NCVOGs database, in addition to BLASTp hits against Cedratviruses and other unclassified viruses, indicating that it is probably more closely affiliated with eukaryotic giant viruses than elements that infect bacteria (as Caudoviricetes). Future studies focused on collecting novel sequences in the same or similar environment may reveal if what we found is a viral genome coming from a true novel chimeric viral group or, an endemic case of chimerism at the Lemonade Creek sites.

-Line 486 – unfortunately IMG/VR does not include an up-to-date database of giant virus genomes. A more complete set that has been decontaminated (i.e. cellular contamination removed) can be found in the GVDB, which also includes a hierarchical taxonomy for the viruses (<https://faylward.github.io/GVDB/>). I don't think this will impact the results here that much, so I don't think that remaking the trees is necessary, but the authors may wish to keep this in mind for future viral studies.

We have now updated the taxonomies of vMAGs affiliated with giant viruses using this useful database, together with environmental source and host prediction of homologous sequences (Figure 4, Extended_Data_Fig1_revision, and Supplementary_table_8).

Congratulations on the exciting study.
Frank Aylward

Thank you very much Frank for all the helpful suggestions and discussions!

Reviewer #2 (Remarks to the Author):

In the manuscript, the authors investigated the viral communities from samples of three distinct features below the tributaries of Lemonade Creek in Yellowstone National Park. This broadly entailed assessment of i) the taxonomic composition of the communities, ii) the likely role of the viral communities in genetic exchange among the different life-forms in these communities, and iii) the potential thermal adaptation of viral genomes in these communities. Overall, this work represents an interesting exploratory effort at a well-restricted, local scale, with unexpected results and rigorous analysis of data. The figures are also very well constructed and of high quality, although some minor suggestions on these are outlined below. Overall, I believe the manuscript would be of interest to a broader community, ranging from biologists working in extreme environments to evolutionary biologists, and shows pretty clearly that viruses should be analyzed with cellular community members, and that large differences in community structure can be found at very localized scales. The manuscript represents great, novel, and exciting science, and my suggestions are mainly associated with potential improvements that the authors may want to consider to the manuscript itself and its structure.

We greatly appreciate the supportive comments from this reviewer.

Some larger comments:

1. Overall, the manuscript did not read as comfortably as it could. The flow of the manuscript felt slightly disjointed, with some large-scale conclusions mentioned in the results in some sections (e.g., line 151-152 "These results suggest that hot springs harbor many novel proteins that may play specific ecological roles in each environment."), while at other times, the context is only provided in the discussion, with no context for the results being discussed in the results section (e.g., section on footprints of extremophily adaptations). These slight differences in writing style read as different contributors having written different sections of the manuscript and overall contribute to a slightly disjointed feel for the reader. I think a potential solution for this would be to combine the results and discussion for the manuscript, as providing the context and implications of the results as they are being discussed may be easier than deconvoluting sections that already contain some discussion in the results.

We appreciate the opportunity to clarify the use of summarizing sentences at the end of each results section. We wrote these closing sentences with the aim of providing the reader a major "take-home message" from each results section. Indeed, the only section which did not include a summarizing sentence was, "Footprints of genome-wide adaptation to extremophily at Lemonade Creek". We now added the following sentence

to this section, in P9 L248 at the end of the section: "These findings indicate that Lemonade Creek vMAGs have genomic signatures of thermophilic adaptation, suggesting a long-term relationship with hot spring environments." However, later in line 220 we agree that we had improperly added a larger-scale conclusion that has now been moved to the discussion P10 L263: "These splits likely occurred when their hosts initially colonized extreme environments, emphasizing the ancient and divergent nature of hot springs vMAGs, when compared to their virus relatives".

2. Furthermore, there are several areas later in the manuscript where the authors refer to analyses to address a particular aim (e.g., line 157 "For this aim,..."), however, these aims weren't clearly defined in the manuscript. Overall, I believe the reader would benefit from having the last paragraph of the introduction refined to have a clearer layout of the aims for the study. This would then also provide a clearer linkage to why certain aspects of the study was undertaken in a particular way. As I outlined above, my take home aims of the manuscript were "i) the taxonomic composition of the communities, ii) the likely role of the viral communities in genetic exchange among the different life-forms in these communities, and iii) the potential thermal adaptation of viral genomes in these communities". As such, if the manuscript was slightly more focused along these three aims explicitly, it would feel much more cohesive and easier to follow for the reader.

We appreciate the reviewer's suggestion of adding aims to the introduction, therefore we modified the end of the introduction (P4 L73) as: "To discriminate between these two hypotheses, we investigated viruses infecting red algal mats in a hot spring environment with the over-arching goals of characterizing viral community composition, elucidating local adaptation and potential role in cellular communities, and understanding virus evolutionary history".

3. Although the authors have provided a lot of useful supplemental information, something that is slightly difficult to assess from the phylogenies provided is the clear link between early diverging lineages in certain viral classes and the rest of the members of those viral classes, and the linkage to thermal/extreme environments. This is because all of the phylogenies are collapsed at various nodes and one cannot really dig into the data to assess the relationships within those groups (e.g., whether obtained relationships reflect previously obtained groupings thus supporting the robustness of the trees/ whether any of the novel taxa fall within collapsed branches etc.). A potential solution for this would be to also include either uncollapsed phylogenies as PDFs in the supplement, or to include the newick files of the actual trees as supplements. Furthermore, something that I would find useful would be mapping environments or thermal or acidic conditions onto the trees for the viruses included in the analyses, or

minimally to Figure 4. I think this would give a greater visual representation of those early diverging viral lineages being mostly extreme environment associated and would give an easier, visual reach to the early Earth environments discussions.

We once again thank the reviewer for these suggestions. We have now provided supplementary figures with all phylogenies un-collapsed and with updated taxonomic annotations, together with environment and host sources (please see Supplementary_table_8 and Extended Data Fig. 1). In addition, the revised Newick files can be found at <https://zenodo.org/doi/10.5281/zenodo.10465700>

4. Additionally, with all of the data presented, I do find the discussion of the significance of the various aspects slightly short, and more analogous to a conclusion section. I realize there are word limits that the authors have to adhere to, but perhaps consider having supplemental text if appropriate to expand on the discussion of various aspects? This would then also afford the authors space to really tailor the results and discussion to hitting those aims hard, and move sections that detract a little from the main story to the supplement (e.g., most information in the section on the vMAGs etc.).

Unfortunately, space constraints were a major consideration when deciding on text content, which influenced the discussion length. Therefore, we did our best to present meaningful discussion and references in a concise fashion to make it easier for our readers to understand the results and their significance.

Minor comments:

Line 46: "in the modern Earth" suggest to change to "on the modern Earth" unless the authors are specifically referring to subsurface extremes.

Thank you, we updated the text.

Line 92: I think this must be a typo for the largest vMAG being 22 Mb? Elsewhere it is indicated as 2.2 Mb, and this is ~10x larger than most prokaryotic thermophile genomes, more on par with some eukaryotic genomes.

It was indeed a typo, which we have corrected.

Line 107-109: Combined with the comment for Fig. 1 (see below), you could change this sentence then to: "Maveriviricetes, although only comprising 0.5% of the total vOTU count, are present in all environments, which is expected given the high abundance of their known viral hosts (Megaviricetes). Although Maveriviricetes only comprise 0.5% of

the total vOTU count, they are present in all environments (*referring to a new supplemental table for specifics*), which is expected."

We changed the sentence to: "Maveriviricetes, although only comprising 0.5% of the total vOTU count, are present in all environments, which is expected given the high abundance of their known viral hosts (Megaviricetes)".

Line 116-120: A little expansion on why the endolythic samples might be so varied compared to the soil or creek samples would help contextualizing the result. Is this a common trend? Are there specific factors influencing the endolythic samples (e.g., distance from spring/rate of dessication etc.) compared to other samples? Etc.

We are not sure why the endolithic samples contained the highest viral diversity, although it is possible that organisms living inside these rocks are better protected from damaging UV irradiation (high in Yellowstone National Park, which has an average elevation of 8,000 ft), but also from elevated temperatures and acidic conditions from the creek. We added this sentence in P5 L120: "This suggests that despite being adjacent, the studied environments contain unique distributions of viral classes, with the endolithic environment harboring a high and non-homogeneous diversity of viruses. This may be explained by endoliths containing unique internal microhabitats which are better protected from low pH, high temperature, and the damaging UV irradiation associated with the other two environments in YNP (i.e., the park has an average elevation of ca. 8,000 ft [<https://www.nps.gov>])".

"Functional profile of the Lemonade Creek virus community" section - the authors state here that the viruses could participate as potential active members in various geochemical cycles, but I am missing clear links that support this claim in text. It is difficult for readers to contextualize these conclusions without clear evidence discussed in text, e.g., specific genes associated with the different cycles being discussed (if not in the main text, at least in supplemental text?).

We agree that we did not expand on this discussion and provide evidence for a causal relationship between genes and the environment, therefore we have omitted this paragraph from the revised manuscript.

Line 195-196 and 203-204: Some abbreviations are not expanded on first use, but only upon subsequent uses, e.g., GVOGs and NCVOGs - please check this carefully.

We have corrected these abbreviations.

For figures:

Overall, the font size of most figures may make it slightly difficult to read the content, thus an overall increase in font size for all the multi-panel figures would be great if possible.

We have increased the font size, when possible, in all figures in the updated files.
Thank you for this comment.

Fig. 1: Perhaps combining all portions of the population that occur in <1% into an "other" category as opposed to listing them individually, as the vast majority of the taxa are not visible in the bar graph, and then including a supplemental table with the actual full proportions/percentages? This would make the panel feel much less cluttered, and clearly highlight those differences among the highly abundant taxa among the samples.

Because we only have 13 virus classes in all environments, we thought that it is important to not omit any virus groups, given that there are not many reports of viruses in hot spring environments, and to avoid grouping in another synthetic category such as "others" in the same way as "unclassified" viruses. Given that the dominant viruses were Megaviricetes, and Caudoviricetes, lower abundance reads did not compromise the overall bar-plot representation we used. In addition, some groups are unique to certain environments and important for discussion in the main text.

Fig. 3c: Showing which domains of cellular life is referred to with the different phyla etc. would greatly help. Thus, ordering the key into discrete cellular domain groups and then adding brackets for indicating where those domains are on the upper axis of the figure would assist in making the figure clearer.

We have added more text to clarify the position of cellular and viral groups in the figure according to the color legends (alphabetically), which we believe are better to follow up in P31 L1002: Sankey diagram showing the cellular sources of HGTs (top) for each virus class (bottom). The key on the bottom shows archeal, bacterial, eukaryotic, and viral taxonomic groups, and is ordered alphabetically in the same direction as the Sankey diagram. The scale bar shows the amount of putative HGT-derived genes.

Fig. 4a: Having environmental context mapped onto this phylogeny would help a ton with visually showing the linkage between the earlier diverging lineages and extreme environments.

We could not find an effective and aesthetically pleasing way to present environmental source in the Figure 4 phylogeny, given the large number of tips that we included to

show the relationships relative to the major viral groups. However, we have now provided un-collapsed and environment mapped phylogenies in the updated Extended Data Fig. 1 file.

Fig. 5 I think expanding panels A and B to stretch the full width of the page, and then having panels C, D, and E below A and B would greatly increase the readability and layout of the figure.

We thank you for this suggestion and we have now provided an updated Figure 5 reflecting these changes.

We once again thank you for all the helpful corrections, suggestions, and discussions.

Reviewer #3 (Remarks to the Author):

In this study the authors explore viruses attuned to geothermal habitats, specifically at Lemonade Creek in Yellowstone National Park. They assessed function (GO annotations and other methods) and acquisition of genes (i.e., HGT) using metagenomic sequencing to search for virus adaptations to these environments where their extremophilic hosts persist. Sampling efforts focused on three environments: (1) creek biofilms produced by Cyanidioschyzon (red alga), adjacent (2) soils and (3) endolithic materials. Their efforts uncover signatures of a “thermophilic lifestyle”, and authors conclude that the virome at these geothermal habitats present ancient adaptations that have been maintained alongside the red algae found at the site (1.5 Bya). Broadly, the authors conclude that their viromes present evidence of the same “underlying rules” of adaptation to these environments that occur among other domains—with practical mention of the physical limitations of proteins (i.e., limited length) and managing folding (i.e., thermostability and the prevention of RNA-RNA interactions) with context to codon bias and other signatures within their data. To meet their conclusions the authors use several well supported programmes for their bioinformatics pipeline/workflow, including the use of viral markers to achieve stringent results. I want to call specific attention to the authors’ dedication to making public their custom scripts—thereby providing transparency and reproducibility of the manuscript and broadly contributing to the community which likely includes others wanting to make use of their workflow. I enjoyed the manuscript and my comments are primarily out of interest/curiosity.

Additionally, this manuscript represents novel findings and identifies viruses of interest from several (putative) groups in an environment largely unexplored. It also makes larger conclusions about the domain of viruses overall in reference to the specific environment—which will be of interest to the greater virus community.

We appreciate these supportive comments about our work.

COMMENTS

(1) Was anything done to enhance the retrieval of viral sequences from the source material, for example DNase application to rid “contaminant DNA” (i.e., nucleic acids outside of viral capsules/non-viral origin DNA). I do not see mention of this in the extraction portion of the methods, however I know JGI in-house processing can include additional steps. Mostly, I am curious about the potential (some) of your identified viruses to be (anciently + degraded) endogenous viruses given the results of Figure 4C where many bins are missing an identifiable MCP (which could be lost or made unidentifiable over time). Given that presence of an MCP gives substantial evidence of a virion stage, and that no or minimal efforts to remove “contaminant” eukaryote or prokaryote DNA could result in sequencing viral inserts from non-viral genomes I am wondering how likely you think this could be the case for some of these bins? Your methods do a great job of removing “contaminants” so I am wondering if viral inserts could be leftover from eukaryotic or prokaryotic genomes.

We did not use a virus enrichment approach to enhance their retrieval from the bulk metagenome data, rather, we screened these data using a stringent virus identification bioinformatic pipeline. As this reviewer recognizes, our pipeline to target high confidence viral genomes from the scaffolds minimizes the retrieval of “contamination” in the form of cellular degraded genomes or virus integration in hosts, at the cost of having viral scaffolds enriched with true cellular integrations. Therefore, we do not think that we have captured fragments of cellular genomes, at least not in a significant number.

(2) Additionally, I am interested in bin750 given the large number of MCP hits it contains, with the current legend I cannot tell the estimate number of MCPs that were identified (seems to be >8 at least). Although I understand the paper is focused on adaptations to this extreme environment, have you done any work to evaluate potential hosts of these viruses, if bin750 is truly coding for so many MCP it could be reflective of its ability to infect your most abundant eukaryote(s) in the system. Given this, I am also interested in which samples the majority of these sequencing reads bin750 originated from (of the three environments). However, I understand this may be outside the scope of your study. I do feel that broadly the sample origin of the VMAGs (i.e., reads mapping reports) is an important component of the story that is missing given that the three environments are seemingly different (enough)—hopefully I have not simply missed this information somewhere.

Indeed, bin750 showed a high number of MCPs, however we are not sure why this is the case and if it is relevant, given that Mimiviridae viruses usually contain a proportion of paralogous MCPs. In this vMAG, all mapped reads came from the endolithic and soil environments. We did not perform any host linkage analysis because at the time of our study we did not have a complete cellular database to perform such an analysis. The number of each virus marker for each bin can be found in the "Supplementary table 6" NCVOG-GVOG sheet. Given the scope of our study and space constraints, we did not discuss the origins of each bin according to specific environment/sample, however in the revised manuscript we include a new sheet in "Supplementary table 2" (RPKM_values) with the coverage of sequencing reads per viral bin and environment.

(3) Going back to the phylogenetic analyses: ModelFinder test was used within IQTREE, which model(s) were established for the trees? Can you also report the size of the alignment that each tree is based off of? I am asking given the use of --automated in trimAl, which can be quite aggressive on virus data.

We have now provided a new sheet in the revised Supplementary table 8 (model_alignment_info) with the selected evolutionary models and the alignment size used for each marker gene tree. We agree that trimAl can be aggressive, however we do not think this poses a problem for our alignments because we were able to recover the known and established relationships between virus reference groups.

(4) How was sample E1 processed with the biodiversity indices, given that it was amplified? I can see by Figure 1C it is likely contributing to the variation shown on your indices. Do you have some ideas of why E1 is also so different or else why E2 and E3 happen to be so similar compared to the other endolithic samples? Furthermore, how were the 51% of unclassified vOTUs handled within the biodiversity indices given that no class level was achieved. Were these excluded from the analyses or have I misunderstood? More specifically, in Figure 1 are the "unclassified" in 1B represented in the indices of 1C? If so then how? If not then do you think they could contribute in any meaningful way to your results?

Actually, it was sample E4 that was amplified using PCR, however all of the endolithic samples were treated equally. In fact, all endolithic samples showed a similar viral taxonomic composition (with more rare taxa), however their proportions were non-homogenous, and now we include a brief explanation of why this is the case. We think that we capture a real signature from these environments, where the nature of the endolith (inside the rock) offers protection to cellular hosts, therefore we would expect more viruses. We now include these observations in the revised manuscript on P5 L120. Regarding the unclassified viruses, despite totaling 51% of vOTUs, they were not

the most abundant group of viruses, however if we had removed it from the diversity analysis, because of their artificial grouping, this could hide variation between samples that could be explained by viral rarity and uniqueness in each environment. Regarding the unclassified viruses, although they constitute 51% of vOTUs, they were not the most abundant group. We think that excluding them from the diversity analysis could potentially obscure variation between samples attributed to rare and unique viral species in each environment. Therefore, we decided to retain them to ensure a complete representation of viral diversity across all samples.

(5) What are these Circoviridae? CRESS viruses as a group can infect microbes, however Circoviridae specifically are typically associated with metazoans, I am surprised you recovered them with high quality + high completeness. What is your impression of these classifications and their potential hosts in this environment?

Unfortunately, these bins identified as CRESS viruses were not informative in ecological and evolutionary terms, primarily given their low number of genes (characteristic of this larger group), but also because most of the hits were to environmental and not host-affiliated sequences. Therefore, we decided to keep the affiliation in the broad group of circular rep encoded single stranded viruses, given the presence of replicases (REP) in all of them. In the literature, they seem to be associated with eukaryotic hosts, but at least for environmental CRESS sequences, their ecological role(s) are still an open question.

(6) Finally, I think the manuscript needs a bit of context within the complete virosphere. The paper features ss and dsDNA viruses, discusses prokaryotic and eukaryotic viruses, however there is no mention at all about the existence (or potential existence) of RNA viruses in geothermal habitats. Given that the larger context of the paper is a comment of viruses functioning similar to other domains in terms of adaptation to these environments it would be worth it to cover any studies related to RNA viruses at these sources, or a mention of the lack-there-of. Especially given that the biofilms are focused on a eukaryote (red alga), and RNA viruses can play important roles in eukaryotes-or at least infect them. For example, the preprint paper “Distinct groups of RNA viruses associated with thermoacidophilic bacteria” (Urayama et al.; doi.org/10.1101/2023.07.02.547447) is exploring this (albeit in prokaryotes). I say this recognising that capturing RNA viruses requires preplanning, however I think they are a component missing from the paper overall given its larger commentary.

We only found four out of 3,679 vOTUs as being classified as Ribodnaviria (please see Supplementary table 1), therefore, in our study we focused only on DNA viruses. This bias likely reflects our use of a DNA based metagenome data, therefore RNA viruses,

lacking a DNA stage, would not be captured. Indeed, there is a lack of studies focusing on RNA viruses in hot springs environments, but also there may be ecological or physical constraints that challenge the colonization and/or persistence of these groups of viruses in hot springs, such as the lower stability of RNA. Because we recognize the need for more studies targeting RNA viruses in extreme environments, particularly in hot springs, we added a sentence in P11 L306 to highlight this point: Our study identified only 4 out of 3,679 vOTUs mapping to RNA viruses (Riboviria), likely due to a bias favoring DNA viral genome amplification rather than their true absence. Despite the challenges posed by extreme conditions in YNP, such as high heat and acidity that may compromise RNA stability, further exploration of RNA viruses in these mats is necessary to understand their ecology and potential genome adaptations under extreme conditions.

To the authors: thank you for sharing your research.
Sincerely,
E.E. Chase

We are truly grateful for your helpful corrections and suggestions to improve our study.

REVIEWERS' COMMENTS:

Reviewer #1 (Remarks to the Author):

Thank you to the authors for addressing my comments. I think it is easier to interpret the trees with the new taxonomy - it seems like some of the bins are from early-branching Megamimivirinae, which is quite fascinating. Also, the AAI of 40% for the pandoravirus bin compared to other pandoraviruses may seem a bit low, but it is actually relatively high considering the rapid evolutionary rate of these viruses. So this bin will be quite valuable for future work examining the emergence of pandoraviruses. Congrats to the authors on an excellent paper!
Frank Aylward

Reviewer #2 (Remarks to the Author):

Thank you for considering all recommendations and implementing suggestions where appropriate for an already great manuscript. I have not further comments or suggestions. I thoroughly enjoyed your manuscript!

Reviewer #3 (Remarks to the Author):

I wish to thank the reviewers for their responses to my comments and amendments to the paper (based on my feedback and that of the other reviewers).

I have no further comments.